# Characterization of the horse chestnut genome reveals the evolution of aescin and aesculin biosynthesis

Wei Sun[1,2,10], Qinggang Yin [1,3,10], Huihua Wan [1,10], Ranran Gao [1,3,10], Chao Xiong[1,4], Chong Xie[5], Xiangxiao Meng[1], Yaolei Mi[1], Xiaotong Wang [6], Caixia Wang[1], Weiqiang Chen [1], Ziyan Xie[6], Zheyong Xue[6], Hui Yao[7], Peng Sun [1,3], Xuehua Xie[1], Zhigang Hu[8], David R. Nelson[9], Zhichao Xu [6] ✉, Xinxiao Sun [5] ✉ & Shilin Chen [1,2] ✉

Horse chestnut (*Aesculus chinensis*) is an important medicinal tree that contains various bioactive compounds, such as aescin, barrigenol-type triterpenoid saponins (BAT), and aesculin, a glycosylated coumarin. Herein, we report a 470.02 Mb genome assembly and characterize an *Aesculus*-specific whole-genome duplication event, which leads to the formation and duplication of two triterpenoid biosynthesis-related gene clusters (BGCs). We also show that *AcOCS6*, *AcCYP716A278*, *AcCYP716A275*, and *AcCSL1* genes within these two BGCs along with a seed-specific expressed *AcBAHD6* are responsible for the formation of aescin. Furthermore, we identify seven *Aesculus*-originated coumarin glycoside biosynthetic genes and achieve the de novo synthesis of aesculin in *E. coli*. Collinearity analysis shows that the collinear BGC segments can be traced back to early-diverging angiosperms, and the essential gene-encoding enzymes necessary for BAT biosynthesis are recruited before the splitting of *Aesculus*, *Acer*, and *Xanthoceras*. These findings provide insight on the evolution of gene clusters associated with medicinal tree metabolites.

Plants synthesize a wide variety of structurally diverse medicinal compounds, such as noscapine, casbenes, avenacins, cucurbitacins, and taxadiene, which are exclusive to specific lineages or species. Interestingly, the biosynthesis of numerous compounds results from a biosynthesis-related gene cluster (BGC), which is characterized by physically colocalized genes involved in a common metabolite pathway[1–6]. With the growing body of genome sequence information available, multiple investigations have been completed to clarify the collinear BGC organization within and across species[7–9]. Examples include triterpene thalianols in *Arabidopsis thaliana* and its close relatives, benzylisoquinoline alkaloids in *Papaver*, steroidal glycoalkaloids in Solanaceae, triterpene cucurbitacins in Cucurbitaceae, and

[1]Key Laboratory of Beijing for Identification and Safety Evaluation of Chinese Medicine, Institute of Chinese Materia Medica, China Academy of Chinese Medical Sciences, 100700 Beijing, China. [2]Institute of Herbgenomics, Chengdu University of Traditional Chinese Medicine, 611137 Chengdu, China. [3]Artemisinin Research Center, Institute of Chinese Materia Medica, China Academy of Chinese Medical Sciences, 100700 Beijing, China. [4]School of Life Science and Technology, Wuhan Polytechnic University, 430023 Wuhan, China. [5]State Key Laboratory of Chemical Resource Engineering, Beijing University of Chemical Technology, 100029 Beijing, China. [6]College of Life Science, Northeast Forestry University, 150040 Harbin, China. [7]Key Laboratory of Bioactive Substances and Resources Utilization of Chinese Herbal Medicine, Ministry of Education, Institute of Medicinal Plant Development, Chinese Academy of Medical Sciences and Peking Union Medical College, 100193 Beijing, China. [8]College of Pharmacy, Hubei University of Chinese Medicine, 430065 Wuhan, China. [9]Department of Microbiology, Immunology and Biochemistry, University of Tennessee Health Science Center, Memphis, TN 38163, USA. [10]These authors contributed equally: Wei Sun, Qinggang Yin, Huihua Wan, Ranran Gao. ✉e-mail: zcxu@nefu.edu.cn; sunxx@mail.but.edu.cn; slchen@icmm.ac.cn

momilactone diterpenoids in grass[9–13]. Evidence from herbaceous plants showed that complex genetic events, such as whole genome duplication (WGD), tandem duplication, gene relocation, chromosomal inversion, and lateral gene transfer, impact the assembly, maintenance, and diversification of genes that are physically clustered and nonhomologous but functionally coordinated during the evolutionary process[13–17]. In contrast, examples that demonstrate genomic mechanisms leading to the exclusive production of pharmaceuticals in trees are limited and include soapbark, yew, and happy tree[18–20].

*Aesculus* L. in the soapberry family (Sapindaceae) is native to the temperate northern hemisphere[21]. This plant accumulates medicinal triterpenoid saponins (escin Ia and escin Ib, collectively referred to as aescin) and the coumarin compound aesculin, all of which possess medicinal properties[22–25]. Escin Ia belongs to barrigenol-type (BAT) triterpenoid saponin, which is derived from an oleanane (β-amyrin) backbone; this backbone further undergoes region-specific C-16, C-21, C-22, C-24, and C-28 oxygenation to become protoaescigenin, the main aglycone of aescin. Protoaescigenin is further modified by the addition of a glucuronyl moiety at the C-3 position, as well as diglucoside and 2-methylcrotonylation at the C-21 position and acetylation at the C-22 position to form escin Ia[26,27]. Aescin preparations have been administered orally, intravenously (sodium aescinate containing escin Ia and escin Ib), and topically in clinical trials to treat chronic venous insufficiency, edema, and hemorrhoids[24,28,29]. Aesculin, which is 6,7-dihydroxycoumarin 6-*O*-glucoside, along with digitoxin, has been used as a popular eye drop solution to alleviate symptoms, such as asthenopia, eye pain, and diplopia[30–33]. Currently, the biosynthesis of aescin and aesculin in horse chestnut trees, as well as the emergence of the BAT biosynthetic pathway in this lineage, remains uninvestigated.

In this work, to determine the biosynthesis pathway of these compounds, we assembled a chromosome-scale genome of *A. chinensis*. We reveal an ancient whole-genome duplication that occurred ~30.8 million years ago (MYA) in *Aesculus*, which might have contributed to the formation of BAT triterpenoid-related BGCs. Furthermore, we conduct functional verification of the key enzymes involved in the biosynthesis of escin Ia, such as AcOCS6, AcCYP716A278,

AcCYP716A275, AcCSL1, and AcBAHD6, as well as the enzymes crucial for aesculin biosynthesis, namely, Ac4CL1-3, AcF6′H1-2, AcUGT84A56, and AcUGT92G7. In addition, we utilize a comparative genomics-driven approach to clarify the organization and evolution of BAT BCGs across angiosperms.

## Results and discussion
### Metabolite profiling of *A. chinensis* in different tissues and capsule structures
To efficiently screen candidate genes that encode enzymes involved in aescin and aesculin pathways, we performed LC–QQQ–MS and explored the metabolite profiles of protoaescigenin (**1**), escin Ia (**2**), escin Ib (**3**), esculetin (**4**), and aesculin (**5**) in branches, flowers, leaves, pericarps, and seeds (Fig. 1A). The contents of **5** were significantly higher in leaves and flowers than in pericarps and seeds ($p < 0.05$), and the contents of **2** and **3** were the highest in seeds ($p < 0.05$) (Fig. 1B, Supplementary Tables 1 and 2). In parallel, matrix-assisted laser desorption/ionization (MALDI)-mass spectrometry imaging (MSI) was performed to determine the spatial distribution of candidate metabolites in capsules. The analytical results showed that **2** and its isomer **3** ($m/z$ 1169.5152 $[C_{55}H_{86}O_{24} + K]^+$) massively accumulated in the cotyledons near the testas (Fig. 1C, Supplementary Table 3).

### Genome assembly and gene annotation of *A. chinensis*
The size of the *A. chinensis* genome was predicted to be 481.90 Mb using flow cytometry (Supplementary Fig. 1). A genome survey of *A. chinensis* based on 17 $k$-mer frequencies of Illumina short reads showed that the *A. chinensis* genome is 504.28 Mb with a small heterozygous peak and an obvious repetitive peak, suggesting that the genome has a low level of heterozygosity (~0.37%) (Supplementary Fig. 2). In addition, long-read sequencing using Oxford Nanopore Technologies (ONT) obtained 34 Gb of data with ~68× coverage and an N50 length of 11.74 kb (Supplementary Table 4). After error correction, trimming, and assembly, filtered ONT reads were assembled into 656 contigs with a total size of 470.02 Mb and an N50 length of 2.05 Mb, covering 97.50% of the estimated nuclear genome size (Supplementary Table 5).

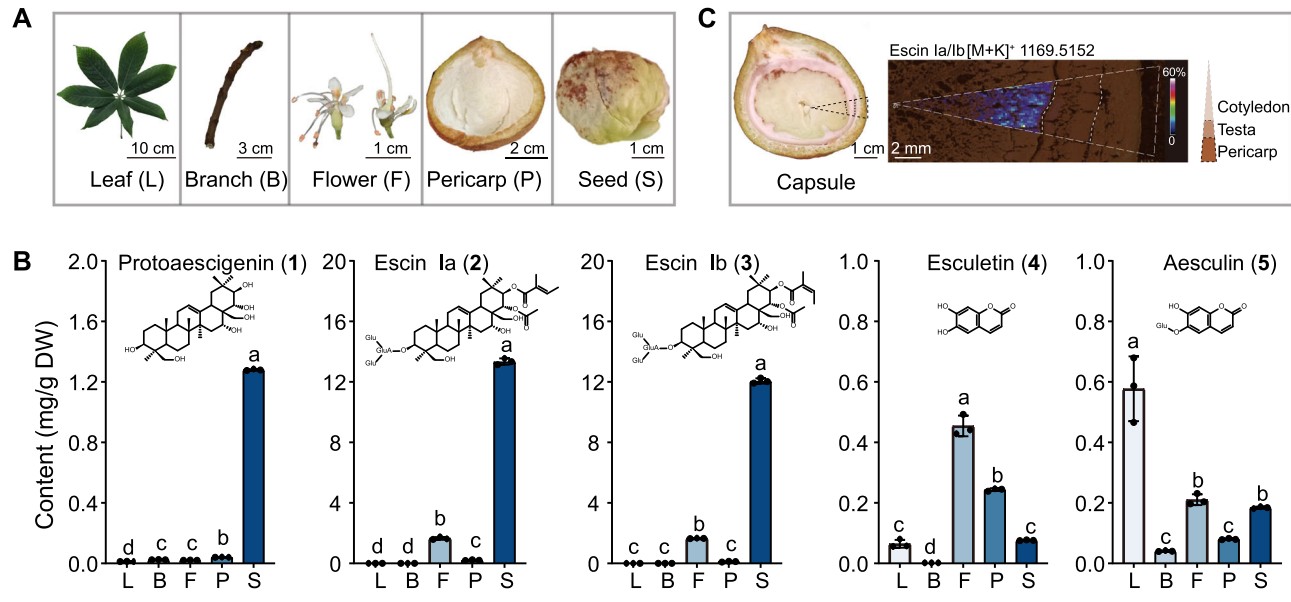

**Fig. 1 | Profiling of aescin and aesculin-related compounds in different organs of *A. chinensis* during development. A** Phenotypes of leaf (L), flower (F), branch (B), pericarp (P), and seed (S). **B** LC–MS analysis shows absolute quantification of protoaescigenin (**1**), escin Ia (**2**), escin Ib (**3**), esculetin (**4**), and aesculin (**5**) in leaf, branch, flower, pericarp, and seed of *A. chinensis*. The data are presented as means values ± s.d. ($n = 3$ biologically independent samples). Different letters represent significant differences ($P < 0.05$) between means according to the analysis of variance (ANOVA) combined with Duncan's multiple range test. **C** MALDI-MSI Image and optimal micrograph of the cross-section of the capsule, including cotyledon, testa, and pericarp. Ion image of **2** or **3** ($m/z$ 1169.5152) in the cotyledon of *A. chinensis* are presented. Source data are provided as a Source Data file.

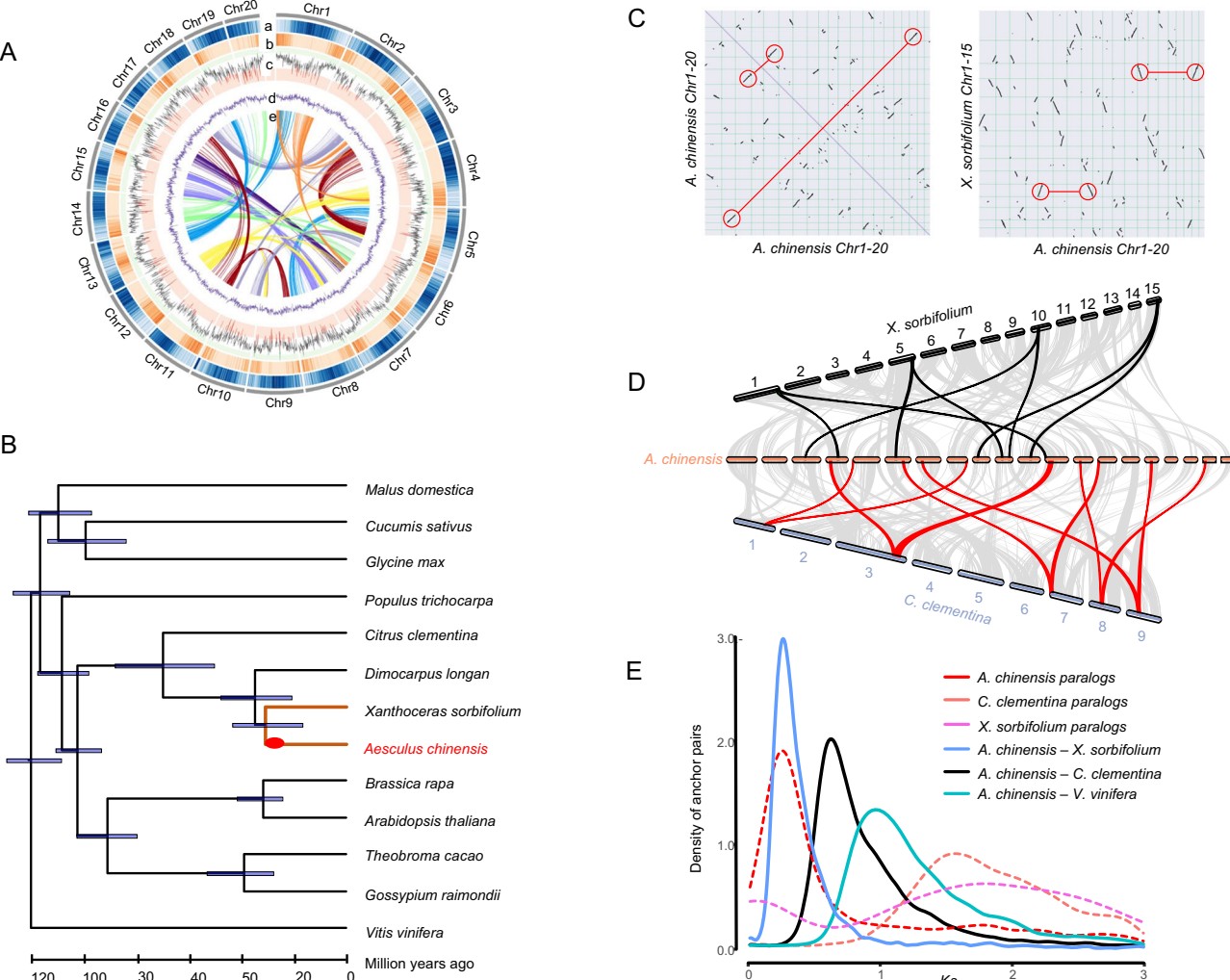

**Fig. 2 | Evolutionary analysis of the *A. chinensis* genome. A** Chromosome-level assembly of the *A. chinensis* genome. a Transposable elements; b gene density; c gene expression level between seed and other tissues; d GC content; and e synteny blocks of paralogous sequences of the *A. chinensis* genome. **B** A phylogenetic tree inferred from orthologs of single gene families among selected plant taxa is shown. The 95% confidence interval for the highest posterior density is shown on the internal nodes. The red circles on the *Aesculus* branch highlight an *Aesculus*-specific WGD event (*Aα*). **C** Macrosynteny between genomic regions of *A. chinensis*, *C. clementina*, and *X. sorbifolium*. Macrosynteny patterns between *A. chinensis* and *C. clementina* or *X. sorbifolium* show that each *C. clementina* or *X. sorbifolium* region aligns with two syntenic regions in *A. chinensis*. **D** Comparison of chromosomes between *A. chinensis*, *X. sorbifolium*, and *C. clementina*. **E** The synonymous substitutions per synonymous site ($K_S$) distributions of orthologous and paralogous genes among *A. chinensis*, *C. clementina*, and *X. sorbifolium*. Source data are provided as a Source Data file.

The assembled contigs comprised many sequencing errors due to the low accuracy of ONT sequencing. Accordingly, this contig-level assembly was further polished three times using Illumina short reads, and 97.4% (1573 out of 1614) of plant single-copy orthologs were identified using Benchmarking Universal Single-Copy Orthologs (BUSCOs) estimation, indicating a high degree of completeness in the polished genome of *A. chinensis* (Supplementary Table 6). In addition, Hi-C chromosome conformation capture sequencing generated 344,838,272 raw paired-end reads, of which 58.05% (189,382,086) were mapped to the contig assembly as unique paired-end reads. Among these unique reads, 174,576,440 were captured to guide the pseudochromosome assembly. A total of 461.08 Mb (98.09%) of the assembled genome was anchored to 20 pseudochromosomes ($2n = 40$) (Fig. 2A, Supplementary Fig. 3). Supplementary Table 5 lists the detailed characteristics of the *A. chinensis* genome.

Approximately 59.14% (276,695,955 bp) of repetitive DNAs in the *A. chinensis* genome (Supplementary Table 5) were annotated in accordance with the transposable element (TE) content of reported Sapindaceae genomes, namely, *Xanthoceras sorbifolium* (56.39%) and *Dimocarpus longan* (52.87%). Of these repetitive elements, 24.37% (114,021,841 bp) of TEs were long terminal repeat (LTR) retrotransposons, of which 98.1% belonged to the *Gypsy* superfamily (32.7%) and *Copia* superfamily (65.4%) (Supplementary Table 7). Moreover, 36,557 protein-coding genes were identified by integrating ab initio gene predictions, homologous protein searches, and the de novo assembled transcripts from RNA-seq reads. In addition, 35,790 (97.9%) genes could be located on the 20 pseudochromosomes. The identified orthologs covered 95.4% of the Embryophyta BUSCOs, indicating that the annotated genome is largely complete (Supplementary Table 8). We further identified orthologous groups of proteomes from *A. chinensis* and 13 other Rosids species, including *A. thaliana*, *Brassica rapa*, *Citrus clementina*, *Cucumis sativus*, *D. longan*, *Glycine max*, *Gossypium raimondii*, *Malus domestica*, *Populus trichocarpa*, *X. sorbifolium*, *Theobroma cacao*, and *Vitis vinifera*, and harvested a total of 21,299 orthologous groups covering 497,631 genes. We then compared the genomes of candidate plant species to obtain gene families that are significantly expanded in *A. chinensis* or unique to *A. chinensis* (Supplementary Fig. 4). Functional prediction showed that the expanded

gene families are especially enriched in the KEGG pathways of secondary metabolites, such as terpenoid biosynthesis (KO00900: terpenoid backbone biosynthesis, KO00909: sesquiterpenoid and triterpenoid biosynthesis) and phenylpropanoid biosynthesis (KO00940) (Supplementary Fig. 5).

## Phylogenomic dating and whole-genome duplication analysis

A total of 139 single-copy genes from 13 Rosids species were selected to construct a high-confidence phylogenetic tree. The phylogenetic trees from concatenated nucleotide and protein sequences of single-copy genes supported that *A. chinensis* is closely related to other sequenced Sapindaceae species, *X. sorbifolium* and *D. longan* (Fig. 2B). In these analyses, *A. chinensis* was found to be a sister lineage to *X. sorbifolium*, and both species further formed a sister group to *D. longan*. Molecular dating of the tested lineages using the nucleotide sequences of single-copy genes and fossil age calibrations revealed that the divergence of Sapindaceae and Rutaceae families of Sapindales occurred at approximately 36.3 MYA with a 95% confidence interval (CI) from the range of 21.8 MYA to 50.4 MYA. The split between *A. chinensis* and *X. sorbifolium* occurred at -32.5 MYA with a 95% CI from 17.5 MYA to 45.4 MYA (Fig. 2B).

Intragenomic collinear analysis identified at least one whole-genome duplication (WGD) event in the *A. chinensis* genome (Fig. 2C). Collinearity analyses between *A. chinensis* and *X. sorbifolium* and between *A. chinensis* and *C. clementina* showed that two paralogous segments in *A. chinensis* corresponded to one orthologous region in *X. sorbifolium* and *C. clementina* (Fig. 2D, Supplementary Fig. 6). These results supported that a species-specific WGD event might have occurred in *A. chinensis* after it diverged from the common ancestor of *A. chinensis* and *X. sorbifolium*. In addition, the distributions of synonymous substitutions per synonymous site ($K_S$) for paralogous genes and anchor pairs in collinear regions of *A. chinensis* showed a clear peak at approximately 0.24 and a minor peak at -1.75, suggesting that the *A. chinensis* genome might have experienced two WGD events (Fig. 2E, Supplementary Fig. 7). Previous studies have suggested that no WGD event occurred in *C. clementina*[34], *X. sorbifolium*[35], and *D. longan*[36] after the ancestral gamma triplication (γ-WGT) event. Our analysis confirmed that only one $K_S$ peak was detected in *C. clementina* and *X. sorbifolium*, at approximately 1.5 and 1.75, respectively. This ancestral $K_S$ peak, shared by the *A. chinensis*, *C. clementina*, and *X. sorbifolium* genomes, represents the γ-WGT event. The $K_S$ distribution of orthologs between *A. chinensis* and *X. sorbifolium* showed one $K_S$ peak at -0.25, slightly larger than the $K_S$ value of paralogs in *A. chinensis*. This result again suggests that the recent WGD event in *A. chinensis* occurred after the divergence between *A. chinensis* and *X. sorbifolium*. Using the divergence time and mean $K_S$ value of orthologs between *A. chinensis* and *X. sorbifolium*, between *A. chinensis* and *C. clementina*, and between *A. chinensis* and *V. vinifera*, we estimated that the species-specific WGD event in *A. chinensis* (Aα) occurred at -30.8 ± 1.33 MYA (Fig. 2B). Furthermore, we identified 3358 genes that retained duplicates from the recent Aα event in *A. chinensis*. The functional annotation and GO enrichment analyses showed that the retained duplicates might be related to the responses of *A. chinensis* to various stimuli (e.g., biotic stimulus, acid chemical, and stress) and the regulation of these responses (Supplementary Data 1).

The origin of large and diverse taxonomic lineages can be related to ancestral polyploidy events[37]. WGD is the major evolutionary force for the production of phenotypic diversity, speciation, and domestication[38]. Examples of WGD-derived plant phenotypic and metabolic diversity that contributes to species-specific gene expansion include oil biosynthesis in wild olive tree domestication[39], ursane triterpene synthesis in loquat (*Eriobotrya japonica*) domestication[40], camptothecin production in happy tree (*Camptotheca acuminata*) domestication[19], and triptolide content in thunder god vine (*Tripterygium wilfordii*)[41]. To investigate the impact of the *A. chinensis*-specific

WGD event on aescin biosynthesis, we systematically calculated $Ks$ for each duplicated paralogous gene pair, emphasizing the upstream pathway genes involved in terpene biosynthesis (i.e., *AACT*, *HMGS*, *HMGR*, *MVA*, *MVK*, *PMK*, *MVD*, *IDI*, *DXS*, *DXR*, *MCT*, *CMK*, *MDS*, *HDS*, *HDR*, *FPS*, *SQS*, and *SQE*) (Supplementary Table 9). We found that the Aα WGD only led to the retention of duplicates in the terpene pathway, suggesting that metabolic flux may have shifted toward triterpenoid metabolism, resulting in aescin production.

## BGC and weighted gene co-expression network analyses (WGCNA) for discovering the escin Ia (2) Pathway

Aescin (e.g., **2** and **3**) are triterpene saponins, biosynthesized from the 30-carbon intermediate 2,3-oxidosqualene (**6**) by the sequential actions of multiple enzymes, including OSC, P450s, UGTs, and acyltransferases (Supplementary Table 10, Supplementary Data 2–4)[42,43]. The pentacyclic triterpene skeleton is derived from a dammarenyl cation and D and E ring expansion en route to β-amyrin (**7**). The **7** further undergoes site-specific oxidation catalyzed by P450s, forming diverse nonglycosylated aglycones, which are collectively referred to as **1**[22]. Acylation and glycosylation of **1** contribute to the structural diversification of triterpene saponins (Supplementary Fig. 8). To identify BGCs involved in triterpenoid biosynthesis in the *A. chinensis* genome, we searched for genomic regions containing OSCs and/or tailoring enzymes (e.g., the most common CYP716 family that modifies triterpene skeletons) known to act in such metabolism. As a result, four BGCs containing OSCs and one BGC containing CYP716 genes were discovered, which are known as AcClusters I–V; these BGCs were then implicated in triterpenoid biosynthesis in the *A. chinensis* genome (Supplementary Fig. 9). Syntenic gene analysis of *A. chinensis* paralogs revealed that AcCluster II is physically syntenic with AcCluster I (Fig. 3A, Supplementary Fig. 9). Of these, AcCluster I is located on chromosome 15 and contains two *OSC* homologs (*AcOSC6* and partial *AcOSC9*), seven P450s (*AcCYP716A274*, *AcCYP716A276*, *AcCYP716A277*, *AcCYP716A278*, *AcCYP716BX1*, *AcCYP716BX3*, and *AcCYP716BX6*), and five *BAHD*s (*AcBAHD1-AcBAHD5*) within a 350-kb region. AcCluster II is located on chromosome 8 and consists of two *P450* genes (*AcCYP716A275* and *AcCYP716BX2*) and one *cellulose synthase-like* gene (*AcCSL1*) (Supplementary Data 5). RNA-seq data further showed that one OSC (*AcOSC6*), four P450s (*AcCYP716A275*, *AcCYP716A278*, *AcCYP716BX1* and *AcCYP716BX2*), and three BAHD genes (*AcBAHD1*, *AcBAHD3*, and *AcBAHD5*) were abundantly expressed in *A. chinensis* seeds, in agreement with the high accumulation of **2** and **3** in the seeds, which putatively participate in aescin formation via the triterpene cyclization, hydroxylation, glucuronidation, and acylation (Fig. 3B, Supplementary Data 5). To expand the scope of candidate genes, further in-depth examination of the WGCNA assay revealed potential genes, including one OSC, three CSLs, nine UGTs, 13 P450s, and 19 BAHDs, in the seed-specific "turquoise" module (Supplementary Fig. 10, Supplementary Data 6). Combining BGCs and WGCNA, we prioritize the validation of highly expressed genes in the seed that are located within the BGCs and subsequently focus on highly expressed genes in the seed that are not present within the BGCs.

## Biochemical identification of genes in escin Ia (2) pathway

Heterologous overexpression of full-length *AcOSC6* (β-amyrin synthase, BAS) in *Nicotiana benthamiana*, which does not naturally generate **7**, led to the production of **7** in plants as measured by GC–MS/MS and confirmed by comparing mass spectral fragmentation with an authentic standard (Fig. 3C, D and Supplementary Fig. 11). The function of AcOSC6 matches its phylogenetic relationships, as AcOSC6 clusters with other OSCs that exhibit BAS activity (Supplementary Fig. 12). To further explore whether seed-specific *CYP716* genes from AcClusters I and II can catalyze the early steps of the biosynthetic pathway of **2**, we co-expressed *AcCYP716A275* or *AcCYP716A278* with *AcOSC6*, as well as *AstHMGR*, to increase metabolic flux to terpenoids using

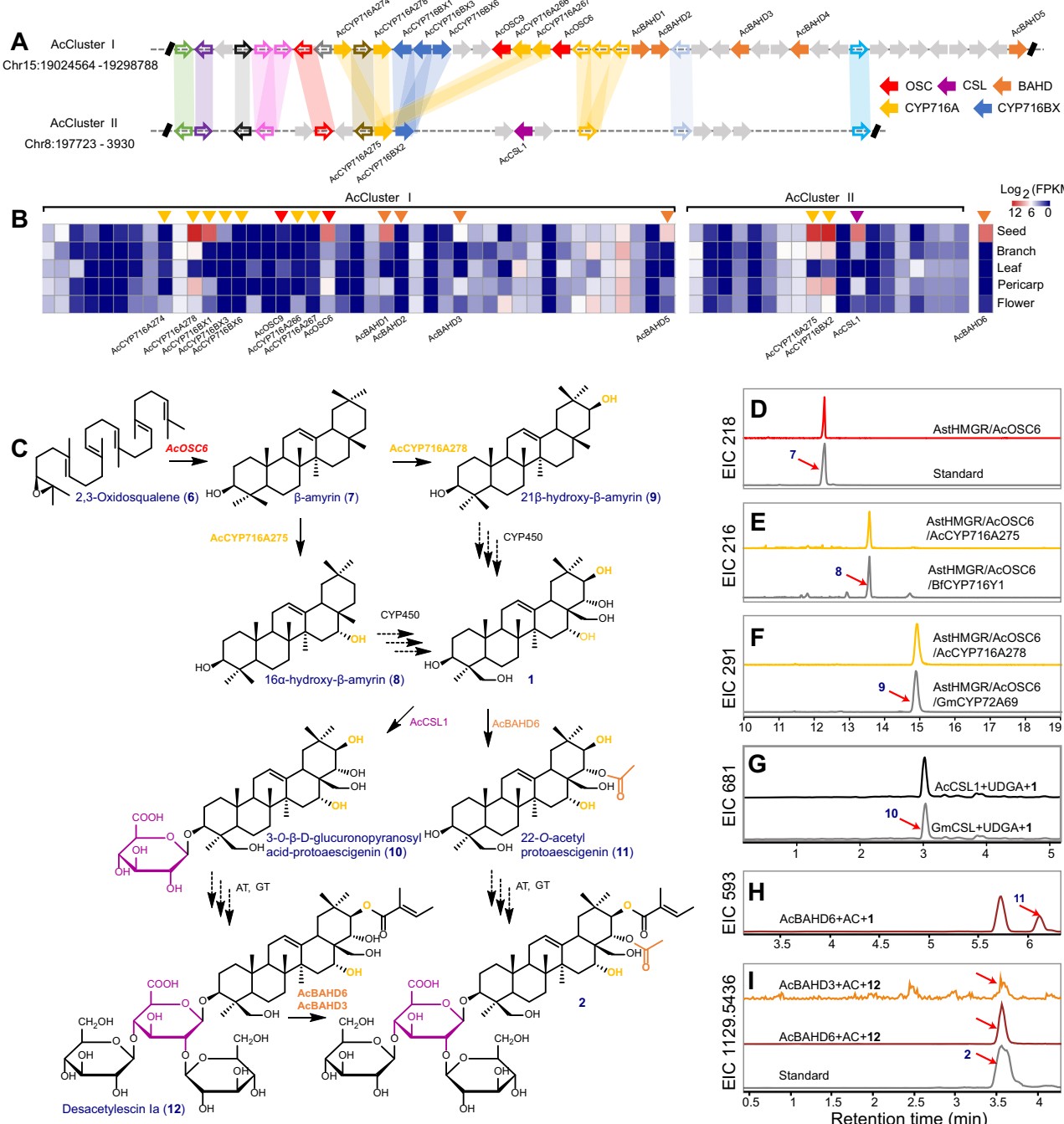

**Fig. 3 | Mining and identification of the escin Ia pathway in *A. chinensis*. A** The collinearity of two BAT BGCs. AcCluster I and AcCluster II present strongly syntenic signals, and the collinearity of these two blocks originated from *Aesculus*-specific *Aα* WGD event. The thickness of syntenic lines represents the identity of syntenic genes. The gray arrows signify the non-syntenic genes. The biosynthetic genes, including *OSC*, *CSL*, *CYP716A*, *CYP716BX*, and *BAHD* members, are marked in different colors. **B** The expression of syntenic genes and *AcBAHD6* related to triterpenoid biosynthesis in different *A. chinensis* tissues, including seed, branch, leaf, pericarp, and flower. **C** The proposed escin Ia biosynthetic pathway and candidate enzymes for each step. Solid arrows represent the transformations verified in this study, and dotted arrows represent proposed reactions. **D–F** Overlay of GC–MS extracted ion chromatograms for β-amyrin (**7**) (*m/z* 218, **D**), 16α-hydroxy-β-amyrin (**8**) (*m/z* 216, **F**), and 21β-hydroxy-β-amyrin (**9**) (*m/z* 291, **E**) produced in *N. benthamiana* expressing *AcOSC6* and *AstHMGR* (KY284573.1), and co-expressing *AcCYP716A275* or *BfCYP716Y1*[44] and *AcCYP716A278* or *GmCYP72A69*[45]. BfCYP716Y1 has been proven to hydroxylate **7** at the C-16α position. Soybean GmCYP72A69 has previously been reported to catalyze the hydroxylation of **7** at the C-21β position[45]. **G, H** Overlay of LC–MS extracted ion chromatograms for 3-*O*-β-D-glucuronopyranosyl acid-protoaescigenin (**10**, *m/z* 681, **G**) produced in yeast WAT11 strain expressing *AcCSL* and *GmCSL*[65] using UDP-glucuronic acid as the sugar donor, and for **11** (*m/z* 593, **H**) and **2** (*m/z* 1129.5436, **I**) produced in *E. coli* expressing *AcBAHD3* and *AcBAHD6*. Soybean GmCSL1 has previously been reported to catalyze 3-*O*-glucuronosylation[65]. Source data are provided as a Source Data file.

*N. benthamiana* infiltration system. Both products exhibited detectable activity toward **7**, creating peaks at 13.6 and 14.9 min, respectively. Co-expression of *AcCYP716A275* and *AcOSC6* in *N. benthamiana* at 13.6 min led to the production of 16α-hydroxy-β-amyrin (**8**), as similarly

verified by comparison to the known production by *AcOSC6/BfCYP716Y1*[44] (Fig. 3C, E and Supplementary Fig. 11). According to the matching MS fragmentation pattern at 14.9 min, the compound produced by *AstHMGR/AcOSC6/AcCYP716A278* is 21β-hydroxy-β-amyrin

(**9**), which is known to be produced by *AstHMGR/AcOSC6/ GmCYP72A69*[45] (Fig. 3C, F and Supplementary Fig. 11). *AcCYP716A275*, *BfCYP716Y1*, *AcCYP716A278*, and *GmCYP72A69* were also, respectively, transferred into engineered yeast strains *Y1-2O-6*, which produces **7**[46], leading to identical catalytic activity as that observed in *N. benthamiana* (Supplementary Figs. 13 and 14). The divergent evolution and multiple functions of CYP72A and CYP716A play crucial roles in the diversity of terpenoids[47–50]. CYP716A subfamily members are involved in catalyzing the conversion of pentacyclic triterpenoids to 28-hydroxy-β-amyrin[51], 16β-hydroxy-β-amyrin[52], 3-oxo pentacyclic triterpenoid[49], and 2α-hydroxy-oleanolic acid[48]. Here, we showed that AcCYP716A275 catalyzes the region-specific C-16α oxidation of **7**, consistent with the broader function of the CYP716A subfamily in triterpenoid oxidation. CYP72A members exhibited diverse hydroxylation activities towards terpenoids, including the 13-hydrolyzation of gibberellins[53], C-30 oxidation of **7**[54,55], C-21β oxidation of soyasapogenol B and **7**[45], C-21β hydroxylation of avenacin triterpenoid backbone[56], C-22β oxidation of 24-hydroxy-β-amyrin[57], and C-23[58] or C-2β oxidation of oleanolic acid[59,60]. Specifically, two CYP72A subfamily members in the CYP72 clan, oat AsCYP72A475, and soybean GmCYP72A69, have been characterized as C-21β triterpene oxidases based on their activity towards oleanane-type triterpenoids. However, we found that a CYP716A subfamily member rather than a CYP72A subfamily member carries out C-21β hydroxylation for the **2** biosynthesis in *A. chinensis*. These results indicate that the C-21β oxidation of oleanane-type triterpenoids underwent independent evolution within the CYP716A and CYP72A (sub)families. Identical compounds produced by distant species may originate from different enzymes; however, these compounds more often arise via the same pathway, regardless of whether these enzymes are homologous[61].

Attaching a hydrophilic carbohydrate fragment to the triterpenoid skeleton enhances its pharmaceutical properties and water solubility[62–64]. Recent reports have identified a series of cellulose synthase-derived glycosyltransferases (CSyGT) that transfer the glucuronic acid moiety to the C-3 position of triterpenoid aglycones in the following leguminous plants: *G. max*, *Glycyrrhiza uralensis*, *Lotus japonicus*, and *Spinacia oleracea*[65,66]. We constructed the phylogenetic relationship of AcCSL1 with characterized and related proteins from different plants available in the literature[65]. AcCSL1 is clustered with a clade of characterized proteins in a CslM subfamily involved in glucuronidation at the C-3 position of oleanane-type triterpenoids, suggesting a similar function in *A. chinensis* (Supplementary Fig. 15). To examine whether AcCSL1 functions as a glucuronic acid transferase, in vivo substrate-feeding, and in vitro yeast assays were performed with the previously characterized *GmCSyGT1* as a positive control. The substrate-feeding experiments, which were performed with recombinant expression in yeast, demonstrated that GmCSyGT1 and AcCSL1 acted on substrate **1**, leading to a peak with a molecular ion at $m/z$ 681.3 Da; this molecular weight is equivalent to that of $[M + COOH]^-$ plus glucuronic acid (Fig. 3G). Enzymatic assays with yeast microsomes containing recombinant AcCSL1 or GmCSyGT1 generated the same results (Supplementary Fig. 16).

Pharmacodynamic studies have shown that acylation at C-21 and C-22 positions increased the cytotoxicity of aescin[67]. A few BAHDs have been identified as major acyltransferases for the acetylation of thalianol-derived tricyclic triterpenoids in Arabidopsis[8,68], tetracyclic cucurbitacins in cucurbits[11], and pentacyclic triterpenoids in spinach and *Boswellia* trees[66,69]. To investigate the activity of transcribed BAHDs (*AcBAHD1*, *AcBAHD3*, and *AcBAHD5*) in AcCluster I and *AcBAHD6* in WGCNA towards substrate **1** with acetyl-CoA as the donor, in vitro enzyme assays were performed using recombinant proteins from *E. coli* (Supplementary Fig. 17). AcBAHD3 and AcBAHD6 could acetylate the hydroxyl group of **1**, yielding a product $m/z$ 593.3 (Fig. 3H, Supplementary Fig. 18). However, the conversion rate of AcBAHD3 was much lower than that of AcBAHD6. The unstable catalytic product of

AcBAHD6 was finally confirmed to be 22-O-acetylprotoaescigenin (**11**) through NMR analyses (Supplementary Fig. 19). Furthermore, AcBAHD3 and AcBAHD6 can use acetyl-CoA as an acetyl donor to catalyze desacetylaescin I (**12**), resulting in the formation of **2** (C22-O-acetylation of **12**). This result was confirmed by comparing the retention time and MS/MS spectra of the product and an authentic standard (Fig. 3I, Supplementary Fig. 20). This study thus provides a second example of a BAHD family member that targets the acylation of pentacyclic triterpenes, complementing a previous report that the BAHD family member BsAT1 from *Boswellia serrata* could catalyze C3a-O-acetylation of α-boswellic acids (BA), β-BA, and 11-keto-β-BA, thus forming all the major C3a-O-acetyl-BAs (3-acetyl-α-BA, 3-acetyl-β-BA, and 3-acetyl-11-keto-β-BA)[70]. Phylogenetic reconstruction further confirmed that the BsAT1 is the closest ortholog of AcBAHD1, 3, 5, and 6, which are placed with the clade IIIa BAHDs representing ATs that act on distinct acceptors (Supplementary Fig. 21). Our results demonstrate that the AcCluster I and AcCluster II BGCs contribute to the biosynthesis of BAT, which we named the BAT BGCs.

## Functional identification of Ac4CLs, AcF6'Hs, and AcUGTs for aesculin (5) biosynthesis

Aesculin (**5**), a coumarin (1,2-benzopyrones), is produced via the phenylpropanoid pathway and distributed in *Aesculus*, *Fraxinus*, and *Populus*[70,71]. The relevant upstream genes identified in this study from the *A. chinensis* genome include seven phenylalanine ammonia-lyases (PALs), three cinnamic acid 4-hydroxylases (C4Hs), five cinnamic acid 3-hydroxylases (C3Hs), four 4CLs, and five F6'Hs (Supplementary Fig. 22, Supplementary Table 11). In this study, three *Ac4CLs* (*Ac4CL1-3*) and four *AcF6'H* (*AcF6'H1-4*) were cloned based on their expression patterns and functionally characterized. Our findings revealed that *Ac4CL1* exhibited a broad expression pattern, whereas *Ac4CL2* and *Ac4CL3* were narrowly expressed in the pericarp. The level of *AcF6'H1* expression was high in the pericarp and branch, but low in the flower, while *AcF6'H2-4* was narrowly expressed in the flower. Enzyme assays using spectrophotometry demonstrated that all three characterized Ac4CL enzymes had a catalytic affinity for caffeic acid (**13**) to be caffeoyl-CoA (**14**) (Fig. 4A, B). Notably, compared to all other Ac4CL enzymes, Ac4CL2 demonstrated superior catalytic properties towards **13** (Supplementary Fig. 23). Enzyme assays showed that both AcF6'H1 and AcF6'H2 exhibited hydroxylation activity with **14** as a substrate. LC-MS/MS analysis of the reaction products, compared with the retention time and MS/MS spectra of the authentic compound, revealed that the hydroxylated products were **4** (Fig. 4A, C, Supplementary Fig. 24). Compared to AcF6'H1, AcF6'H2 exhibited the greatest catalytic efficiency against **14** (Supplementary Fig. 25).

Nineteen putative AcUGTs were identified from phylogenetic analysis based on homology to the known UGTs involved in the flavonoid and coumarin biosynthesis (Supplementary Table 12). These AcUGTs were cloned and expressed in *E. coli* to produce recombinant proteins; the enzymatic activities of these proteins were then investigated using **4** as acceptors and UDP-glucose as a donor substrate (Supplementary Fig. 26). AcUGT84 A56 and AcUGT92G7 were found to utilize **4** (Fig. 4A, D). MS data revealed that both AcUGTs could catalyze the addition of one glucose, as indicated by increases in the molecular weight of products relative to the corresponding aglycones by 162 Da. By comparing the mass spectra and retention time to those of authentic standards, the enzymatic products of AcUGTs with **4** as the substrate were confirmed to be **5** (Supplementary Fig. 27). Enzymatic kinetic tests showed that the affinity of AcUGT92G7 ($K_M = 48.96\,\mu M$) was stronger than that of AcUGT84A56 ($K_M = 177.25\,\mu M$) for **4** (Supplementary Table 13).

## De novo production of aesculin (5) in *E. coli*

The characterized enzymes Ac4CL2, AcF6'H1, and AcUGT92G7 were used to assemble a de novo biosynthetic pathway for compound **5**. As previously demonstrated, the supply of phosphoenolpyruvate (PEP)

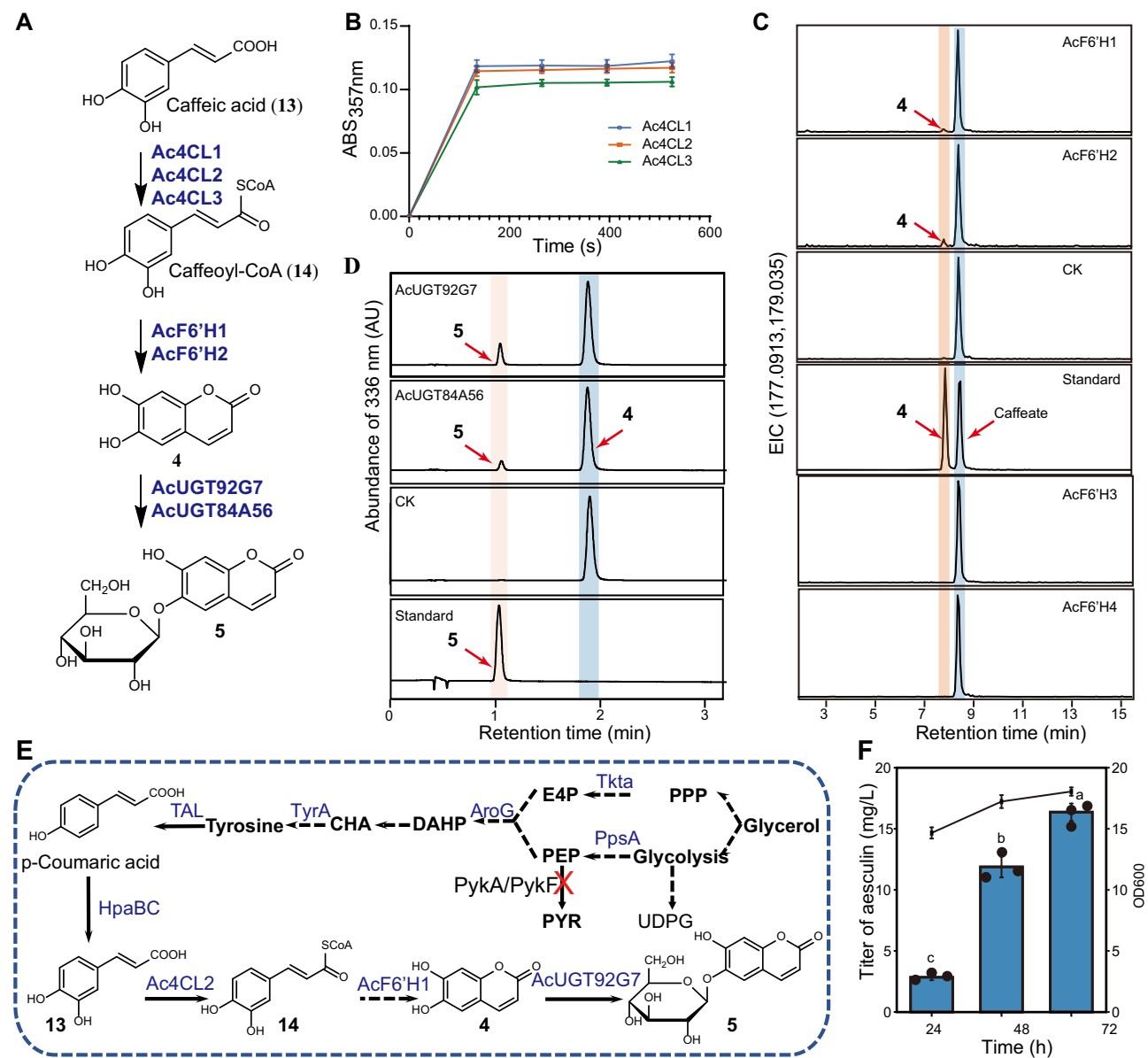

**Fig. 4 | Functional identification of Ac4CLs, AcF6'Hs and AcUGTs involved in aesculin (5), and de novo biosynthesis of 5. A** Enzymes characterized for the biosynthetic pathway of **5**. **B** Ac4CL enzymes activity assay. Ac4CL1-3 isoforms were assayed for activity against caffeic acid (**13**) substrate. Enzyme activity was measured by recording the absorbance at 375 nm of caffeoyl-CoA (**14**) product formation over time. Greater absorbance indicates higher Ac4CL enzyme activity. The data are presented as means values ± s.d. (*n* = 3 biologically independent samples). **C** AcF6'H enzymes activity assay. *E. coli* cells expressing either AcF6'H1 or AcF6'H2 converted substrate **14** into product **4**. LC–MS spectrometry detected a peak with the same retention time and mass spectrum as the standard of **4**. Standard: **4** and caffeate; CK: *E. coli* carrying the empty vector. **D** UPLC analysis for AcUGT92G7 or AcUGT84A56 enzyme activity toward **4**; CK: *E. coli* carrying the empty vector. **E** The artificial pathway for the biosynthesis of **5** from tyrosine in *E. coli*. Dashed arrows represent multistep conversion. The red cross line represents the pathway block;

genes in blue indicate overexpression. PPP pentose phosphate pathway; UDPG UDP-α-D-glucose; E4P D-erythrose-4-phosphate; PEP phosphoenolpyruvate; PYR pyruvate, DAHP 3-deoxy-arabino-heptulonate-7-phosphate, CHA chorismate, Tkta transketolase, PpsA phosphoenolpyruvate synthase, PykA/pykF pyruvate kinase, AroG phospho-2-dehydro-3-deoxyheptonate aldolase, TyrA chorismate mutase-prephenate dehydrogenase, TAL tyrosine ammonia lyase, HpaBC 4-hydroxyphenylacetic acid 3-hydroxylase, 4CL1 4-coumarate-CoA ligase, F6'H feruloyl-CoA 6'-hydroxylase, AcUGT84A56 and AcUGT92G7 glucosyltransferases. **F** De novo biosynthesis of **5** by strain XC-6 (BW-1, pCS-TPTA-HpaBC, and pET-648T). The data are presented as means values ± s.d. (*n* = 3 biologically independent samples). Different letters (a–c) above columns represent significant differences between samples using one-way ANOVA at *P* < 0.05. Source data are provided as a Source Data file.

and erythrose 4-phosphate can be increased by over-expressing *ppsA* and *tktA*, respectively[72]; in addition, tyrosine production can be improved by over-expressing feedback inhibition-resistant variants of *tyrA* and *aroG*[73]. In this study, four key enzymes were expressed to enhance the carbon flux in the shikimate pathway.

For de novo synthesis of **5**, we chose BWΔ*pykA*Δ*pykF* as the basal strain to reduce the conversion of PEP to pyruvate[74]. BWΔ*pykA*Δ*pykF* was co-transformed with plasmids pZE-649T and pCS-TPTA-HpaBC,

forming strain BW1. M9 medium supplemented with 5 g/L yeast extract was employed for de novo production of **5**. The corresponding fermentation curves and titer of **5** are shown in Fig. 4D. The strain growth rate was faster during fermentation for 0–24 h and slowed down after 24 h, and the biomass reached an OD$_{600}$ of 18.1 at 72 h. The production of **5** also increased steadily as fermentation proceeded. The final production of **5** at 72 h reached 16.3 ± 0.7 mg/L. Therefore, we reconstructed a biosynthetic pathway for **5** by combining three genes from

*A. chinensis*. Additionally, these pathway engineering strategies could provide powerful tools and insights to further progress and discovery in coumarin biosynthesis.

## Evolution and organization of BAT BGCs

The origin of plant BGCs stems from the assembly of genes after gene duplication, neofunctionalization, and genomic relocation[75]. The current genome available is abundant and can provide crucial clues into how genes that encode enzymes from a common biosynthetic pathway are assembled into a formation of BGCs[9]. Comparative genomics in combination with functional characterization indicates that early-arising BGCs underwent dynamic changes in auxiliary enzymes to generate structurally diverse triterpenoids within and between Arabidopsis or between cucurbits[75]. However, little is known about the evolutionary trajectories of triterpenoid BGCs in other plant clades.

Our functional data demonstrated that *AcOSC6, CYP716, BAHD IIIa*, and *CSL* genes distributed in AcCluster I and AcCluster II are crucial enzymes for the catalysis of BAT biosynthesis. The mean $K_S$ value of 17 paralogous gene pairs localized in the syntenic blocks of AcCluster I and AcCluster II was 0.27, which is more generally similar to the $K_S$ value of the paralogs in *A. chinensis* (0.24), suggesting that the duplication event of AcCluster I and AcCluster II might originate from the *A. chinensis*-specific WGD event, Aα WGD. Syntenic analysis showed that AcCluster I and AcCluster II are conserved among Hippocastanoideae species, including *A. chinensis, Acer yangbiense*, and *X. sorbifolium* (Fig. 5A), in accordance with the high accumulation of BAT in these species[76–78]. The *CYP716* genes, which are specifically enriched in *A. chinensis, A. yangbiense*, and *X. sorbifolium*, are presumably responsible for the diversity of BAT biosynthesis in these species.

Further large-scale analysis of evolutionary dynamics for this BAT BGC among 21 published plant genomes, including early-diverging angiosperm, monocot, and eudicot genomes, was conducted to examine how this syntenic region may have evolved (Fig. 5A). An ~1-Mb segment of 64 genes (Fig. 5B, C), including one cycloartenol synthase (CAS) gene and one *BAHD IIIb* gene in the early-diverging angiosperm *Amborella trichopoda*, was traced as the ancestral region. Syntenic regions are also present in monocots (*Oryza sativa* and *Zostera marina*), but these lineages only contain one *CAS* gene, not the BAT biosynthesis-related *BAS, CYP716, CSL*, and *BAHD* genes. The *CYP716A* member first appeared in the corresponding region of species within Ranunculales, i.e., *Papaver somniferum*. However, the syntenic region is not present in all examined Superasterid species. After the split with Superasterids species, almost all the syntenic segments retain at least one *OSC* gene except for Cruciferae representatives, such as *A. thaliana* and *B. rapa* in Superrosids. Remarkably, a sizeable species-specific tandem duplication of 10 complete and 10 partial *OSC* genes was found in the conserved syntenic region of the *V. vinifera* genome. Complex insertion, deletion, duplication, and arrangements of *OSC, CYP716, BAHD*, and *CSL* genes in the syntenic regions of Superroside species were observed. The *BAS/CYP716/BAHD* BGC was always found in the syntenic region of Superrosid species, but dynamic duplication and reorganization occurred (Fig. 5A). We propose that the complete BAT BGC *BAS/CYP716/BAHD/CSL* was first assembled in Hippocastanoideae species, i.e., BAT-producing plants (*A. chinensis, A. yangbiense*, and *X. sorbifolium*). In addition, the BGC underwent further dynamic evolution during speciation, such as the tandem gene duplication of *CYP716A* and *CYP716BX* and the *Aesculus*-specific duplication of AcCluster I and AcCluster II from the Aα WGD event observed here.

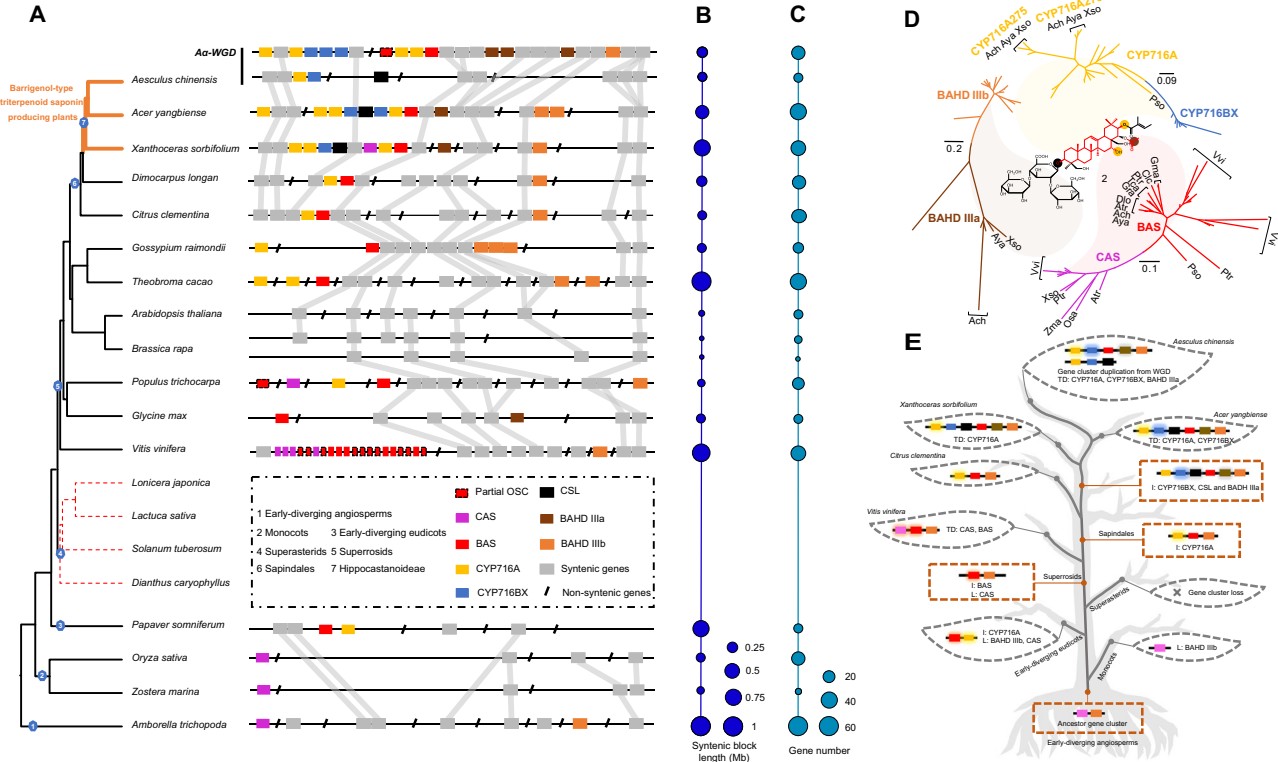

**Fig. 5 | Evolution of BAT triterpenoid biosynthesis-related gene clusters among angiosperms. A** The phylogenetic tree was constructed using OrthoFinder based on the reported genome information. Synthetic genes or non-synthetic genes were marked by different colors, and the ancestral collinearity gene cluster was first identified in the early-diverging angiosperm *A. trichoppda*. **B** The length distribution of syntenic blocks among angiosperms. **C** The gene numbers localized in syntenic blocks among angiosperms. **D** The phylogenetic relationship of OSCs,

P450s, and BAHDs, which are localized in the BAT BGCs. The different colors represent the correspondingly catalytic activities in the biosynthesis of **2**. **E** The evolutionary model of BAT BGCs following many tandem duplications, gene insertions, gene losses, and WGDs. The model illustrates that the BAT biosynthetic genes were swapped and duplicated into the gene cluster and subsequently evolved the accumulation of species-enriched BAT triterpenoids. I insertion, L loss, TD tandem gene duplication. Source data are provided as a Source Data file.

To further examine the evolutionary origins of *OSCs*, *P450s*, *CSLs*, and *BAHDs*, we generated a series of phylogenetic trees using the maximum-likelihood method; this procedure was based on multiple alignments of protein sequences from the aforementioned species for synteny analysis (Fig. 5D). The phylogenetic trees based on the syntenic regions of Sapindales species, including *A. chinensis*, *A. yangbiense*, *X. sorbifolium*, *D. longan*, and *C. clementina*, along with *V. vinifera*, confirmed that *BAS*, *CYP716*, *CSL*, and *BAHD III* genes were conserved in relation to triterpenoid biosynthesis. In addition, gene copy numbers and phylogenetic relationships revealed that the species-specific tandem duplication events of *CYP716* family members in *A. chinensis* and *A. yangbiense* led to the expansion of *CYP716A* and *CYP716BX* subfamily members. The phylogenetic tree of kingdom-wide identification of cellulose-synthase superfamily genes suggested that *AcCSL* genes and their collinear genes from *A. yangbiense* and *X. sorbifolium* as well as the functionally verified *CSLs*, which catalyze the conjugation of glucuronic acid to the triterpenoid backbone, share a close phylogenetic relationship (sequence identities of 44.7–55.2%). These results indicated that *CSL* genes localized in the BAT BCGs of *A. chinensis*, *A. yangbiense*, and *X. sorbifolium* were recruited to catalyze glucuronidation activity before their speciation.

Here, we proposed evolutionary trajectories for the birth, death, and evolution of the BAT BGC among angiosperms based on ancestral state reconstruction (Supplementary Fig. 28). (1) Birth in early-diverging angiosperms: The ancestor BGC contains one *CAS* and one *BAHD IIIb* in *A. trichopoda*; (2) Death in monocots: The region containing CAS and other non-BAT functional genes is retained, but *BAHD IIIb* is lost; (3) Evolution in specific early-diverging eudicots: *BAS* undergoes evolution and *CYP716* is added; (4) Death in Superasterids: The BGC is lost in this lineage; (5) Assembly of the BAT BGC in Hippocastanoideae: This functional BGC (*BAS/CYP716/CSL/BAHD*) is formed, leading to BAT production; (6) Reorganization and diversification of the BAT BGC: This BGC undergoes dynamic evolution in a species-specific manner via tandem duplication and WGD (Fig. 5E).

While the principles of evolution of metabolic pathways with multiple steps are less clear in plants, it is well understood that individual enzymes have evolved through gene duplications to acquire metabolic diversity[79–81]. Clues to determine the underlying causes of compound origin are provided by plant genes linked in the genomic region known as the BGC, which includes various types of encoding enzymes[82]. Currently, many researchers utilize the rich resources of genomic sequences to uncover complex evolutionary procedures that impact the conservation and flexibility of BGC, which determine the appearance, maintenance, and innovation of chemical components in the scope of interspecies or intraspecies[14,83–85]. In this study, our genome-driven tactics provided perspectives on the synthesis and evolution of barrigenol-type triterpenoids across angiosperms. The evolutionary trajectories of BGCs among angiosperms revealed that the functional BGC leading to BAT biosynthesis was specifically assembled in Hippocastanoideae species and that the dynamic evolution was species-specific via tandem duplication and WGD after the speciation of *A. chinensis*. Furthermore, without wild *Aesculus*, it is difficult to economically synthesize aescin and aesculin. Although synthetic biology offers a promising method for producing desired components in heterologous systems, enzymatic reactions responsible for their biosynthesis remain unexploited. Exquisite genome editing techniques might be used to bioengineer crop and microbial species to manipulate the multiple enzymes involved in the aescin and aesculin biosynthesis processes, through which high-value pharmaceutical components can be manufactured on a large-scale.

## Methods
### Plant materials and chemicals
*A. chinensis* plants were collected from the Institute of Medicinal Plant Development (IMPLAD), the Chinese Academy of Medical Sciences

(CAMS), China. *N. benthamiana* was cultivated for a duration of ~4 weeks under an 18 h light/6 h dark cycle within a greenhouse located at the National TCM Gene Bank, Institute of Chinese Materia Medica, China Academy of Chinese Medical Sciences. High-quality standards for protoaescigenin (**1**), desacetylaescin I (**12**), escin Ia (**2**), escin Ib (**3**), esculetin (**4**), and aesculin (**5**) were purchased from Chenguang Biotechnology Inc. (Xi'an, China). Caffeic acid (**13**), disodium 2-oxoglutarate, caffeoyl-CoA (**14**) were purchased from Chengdu Push Biotechnology Co., Ltd (Chengdu, China). UDP-Glucose (UDPG), UDP-Glucuronic acid (UDPGA), adenosine 5′-triphosphate, and S-adenosylmethionine (SAM), acetyl-CoA were purchased from Sigma-Aldrich (USA). All chemicals used were of analytical or HPLC grade.

### MALDI MS imaging
Fresh *A. chinensis* capsules were harvested and promptly encased in a 10% gelatin (w/v) solution. Initially, the tissues were positioned within Tissue-Tek cryomolds (25 × 20 × 5 mm) and enveloped by the gelatin solution. Then, the molds were transferred to a −80 °C freezer for 1 h to solidify into blocks. During cryo-sectioning, the sample blocks were affixed directly onto the sample holder of a cryostat (Leica, Germany) using deionized water as an adhesive. Sections with a thickness of 16 μm were obtained at −20 °C and immediately affixed onto indium tin oxide-coated glass slides for subsequent imaging analyses. To preclude condensation, the tissue sections underwent a vacuum desiccation process for ~10 min before the matrix was applied. Optical images of the sections were captured utilizing a Zeiss Axio M2 microscope (Zeiss, Germany). For consistent matrix application, a custom-designed automated pneumatic-assisted system was employed. The matrix application system and coating procedure were conducted[86]. In brief, a solution of 50 mg/mL 2,5-dihydroxybenzoic acid dissolved in acetonitrile:$H_2O$ (0.1% TFA, trifluoroacetic acid) (7:3, v/v) was applied for positive mode MALDI experiments. The MALDI system (TransMIT GmbH, Giessen, Germany) was operated under atmospheric pressure and was coupled with a Q Exactive HF Orbitrap mass spectrometer (Thermo Fisher Scientific, Bremen, Germany). To ensure uniform deposition onto the capsule samples, the nebulizer was situated 3 cm above the sample and oscillated over the Plate 100 times. The flow rate was set at 6–8 mL/h, while the gas pressure was adjusted to 50 psi to facilitate the delivery and nebulization of the matrix solution.

Each single-scan spectrum was composed of 100 accumulations of laser pulses at a frequency of 1 kHz, employing a "small" laser focus configuration. MALDI images were acquired with a spatial resolution of 50 μm. The instruments were calibrated utilizing the ESI inlet, employing sodium trifluoroacetate (NaTFA) clusters spanning the *m/z* range of 158.9640–1926.6365. Instrument calibration was performed using the ESI portion of the instrument with NaTFA clusters and a quadratic calibration of the clusters. Data analysis was executed utilizing the SMALDIControl software package (TransMIT GmbH, Giessen, Germany).

### Metabolite profiles in *A. chinensis* by LC−MS/MS
All tissues were subjected to vacuum freeze-drying, and each individual sample was precisely weighed to 100 mg and subsequently pooled for subsequent solvent extraction. To effectuate this process, 10 mL of a methanol:$H_2O$ mixture (7:3, v/v) was introduced into 50 mL centrifuge tubes, followed by a 30 min sonication period utilizing an 8891 ultrasonic cleaner (Billerica, MA). Subsequently, the extracts were centrifuged at 14,000 × *g* for 5 min, and the supernatants were collected. These extract solutions were employed for subsequent analyses. For the quantification of **1, 2, 3, 4,** and **5**, LC−QQQ−MS/MS assays were executed utilizing a 6470 Triple Quadrupole mass spectrometer (Agilent Technologies, Santa Clara, CA, USA) coupled with an Advance 1290 UHPLC system (Agilent Technologies, Santa Clara, CA, USA). Analytes were resolved using an Agilent Eclipse Plus C18 column

(RRHD 1.8 µm, 2.1 mm × 50 mm). Mobile phase A comprised 0.1% acetic acid, while phase B was composed of $H_2O$ containing 0.1% acetic acid. The gradient elution program was as follows: 0–25 min, A:B = 68:32. A 1 µL volume was injected, and the flow rate was set at 0.30 mL/min, with the column temperature maintained at 30 °C. The mass spectrometer, which was operated in either positive or negative ion mode, was configured for multiple reaction monitoring (MRM) mode. For each compound, two precursor-product ion MRM transitions were selectively monitored. Data acquisition and subsequent analysis were executed employing MassHunter software (Agilent Technologies, Santa Clara, CA, USA), facilitating the quantification of all seven metabolites for which corresponding standards were available.

### Genome sequencing and assembly

Flow cytometry was employed to estimate the genome size of *A. chinensis* using *Solanum lycopersicum* (900 Mb) and *Brassica rapa* (485 Mb) as the standards on a BD Accuri™ C6 (BD Biosciences, San Jose, CA, USA). In addition, the genome size of *A. chinensis* was also estimated according to *k*-mer frequency (*k*-mer length 17) of genome sequencing data using Jellyfish (v.2.0).

DNA isolation from *A. chinensis* leaves was executed through a plant DNA extraction kit (Tiangen Co. Ltd., http://www.tiangen.com). Short-read paired-end libraries were constructed following the manufacturer's guidelines and sequenced on an Illumina Hiseq X-ten platform (Illumina, San Diego, CA, USA). For long-read sequencing, high-quality *A. chinensis* genomic DNA was employed for the preparation of 20 kb insert libraries, adhering to standard ONT library preparation protocols. These libraries were then sequenced utilizing the ONT GridION X5 platform (v.9.4.1; Oxford Nanopore Technologies). Base-calling of the raw ONT reads was accomplished using the Oxford Nanopore base caller Guppy (v.1.8.5) with default parameters. The ONT reads were subjected to correction, trimming, and assembly into genomic contigs using CANU (v.1.5)[87] and SMARTdenovo[88]. The assembled contigs were further polished three times utilizing Illumina short reads using Pilon (https://github.com/broadinstitute/pilon, v.1.24). The completeness of the genome assembly was estimated using Benchmarking Universal Single-Copy Orthologs (BUSCO; v4) with Embryophya odb10 dataset. A Hi-C library of young *A. chinensis* leaves was prepared by Annoroad Genomics (http://en.annoroad.com) following the standard procedure[89]. The Hi-C sequencing data was aligned to the assembled contigs by BWA-MEM (v.0.7.17), and then the contigs were clustered onto chromosomes with 3D-DNA.

### Transcript sequencing and WGCNA analysis

For RNA-Seq analysis, *A. chinensis* RNA samples were meticulously extracted from diverse tissues, encompassing young leaves, branches, flowers, pericarps, and seeds, using the RNeasy Plant Mini Kit (Qiagen, https://www.qiagen.com). These RNA samples were then employed to construct RNA-Seq libraries utilizing the TruSeq RNA Library Prep Kit v.2 (Illumina, San Diego, CA, USA). Subsequently, the libraries were subjected to sequencing using the NextSeq 500 platform. The quantification of transcript abundance was gauged by calculating the Fragments per Kilobase of the exon model per Million mapped reads (FPKM) values. This quantification was executed through the utilization of HISAT2 (v.2.0.5) alignment and FeatureCounts (v.1.6.3). Employing the FPKM values derived from different tissues, a weighted gene co-expression network analysis (WGCNA) was performed to infer networks of co-expressed genes. Module eigengene values were computed within the WGCNA framework and subsequently correlated with the accumulation profiles of potential metabolites across distinct plant tissues.

### Genome annotation and candidate gene identification

Repetitive sequences within the *A. chinensis* genome were explored through de novo-based and homology-based strategies, facilitated by RepeatModeler (v.1.0.9) software. For de novo repeat identification and classification, the LTR_Finder and LTR_retriever programs were employed to identify and classify repeat elements. De novo transcript assembly was performed using Trinity (v. 2.2.0)[90] with default parameters, and peptide sequences were predicted by TransDecoder (https://github.com/TransDecoder, v.2.1.0). The MAKER annotation pipeline was used for ab initio predictions of protein-coding genes in the masked *A. chinensis* genome (v.2.31.9)[91].

BLASTP-based and HMMER annotations in combination were used to find high-confidence proteins associated with triterpenoid and coumarin biosynthesis in *A. chinensis*. Protein sequences from characterized proteins in triterpenoids and coumarins were used to identify candidate genes by BLASTP (*E*-value < 1e−5, identity > 50%, and coverage >50%) in the *A. chinensis* genome. The identified candidate genes were verified by HMMER; only proteins with representative domains were retained.

### Phylogenetic tree and evolutionary analysis

A maximum-likelihood (ML) phylogenetic tree of single-copy genes, which were clustered into orthologous groups, from the genomes of *A. chinensis* with *M. domestica*, *C. sativus*, *G. max*, *P. trichocarpa*, *C. clementina*, *D. longan*, *X. sorbifolium*, *B. rapa*, *A. thaliana*, *T. cacao*, *G. raimondii*, and *V. vinifera* was constructed using RaxML (v.8.2.10)[92] using the JTT + G + I substitution model for amino acids with 1000 bootstrap replicates. The divergence time of selected species was calculated using the program MCMCTREE in PAML, according to fossil divergence time points for the specific Sapindaceae and Rutaceae (21.8–50.4 MYA) split obtained from TimeTree (http://www.timetree.org). Then, CAFE (v.2.1) was utilized to identify contraction and expansion analyses of these gene families following species divergence prediction under a probabilistic graphical model. MCScan (Python version) was applied to perform genome synteny analysis.

### Gene cluster involved in triterpenoid biosynthesis

We used the plantiSMASH online pipeline (http://plantismash.secondarymetabolites.org/) and a manual search to annotate the triterpenoid BGCs. To assign BGCs to triterpenoid biosynthesis, we surveyed whether these genes contained one or more *OSC* genes and annotated the 20 surrounding genes that were physically adherent. The obtained BGC containing *OSC6* was mapped to the self-genome and four other genomes (*A. yangbiense*, *C. clementina*, *D. longan*, and *X. sorbifolium*) to uncover collinear segments using MCScan (python version).

### Gene cloning

Total RNA was extracted using an EasyPure® Plant RNA Kit (TransGen, Beijing, China). First-strand cDNA was synthesized from total RNA using the TransScript® One-Step gDNA Removal and cDNA Synthesis SuperMix cDNA synthesis kit (TransGen, Beijing, China) according to the manufacturer's protocol. Full-length ORFs of *AcOSC6*, *AcCYP716A275*, *AcCYP716A278*, *AcCSL1*, *AcBAHDs* (*AcBAHD1*, *AcBAHD3*, *AcBAHD5*, and *AcBAHD6*), 19 *AcUGT*s and *AcF6′Hs* (*AcF6′H1*, *AcF6′H2*, *AcF6′H3*, and *AcF6′H4*), and three *Ac4CLs* (*Ac4CL1*, *Ac4CL2*, and *Ac4CL3*) were amplified using KOD-Plus-Neo DNA polymerase (TOYOBO Bio, Japan), and the primers were listed (Supplementary Tables 12, 14 and 15). *AtHMGR* (KY284573), *GmCYP72A69* (NM_001354946), *BfCYP716Y1* (KC963423.1) and *GmCSyGT1* (or GmCSL1, LC500227.1) were synthesized (GenScript Biotech).

### Functional characterization of *AcOSC6* and *P450* genes in *N. benthamiana*

*AcOSC6* and *P450* genes were first recombined into the donor vector pDONR207 and then subcloned and inserted into pEAQ-HT-DEST1 using the LR clonase II enzyme (Thermo Fisher Scientific, USA). pEAQ-HT-DEST constructs were electrotransformed into the *Agrobacterium*

*tumefaciens* (EHA105) strain. Transformed *A. tumefaciens* strains carrying the candidate genes were infiltrated solely or equally in combination with AtHMGR into *N. benthamiana* using a syringe. Infiltrated *N. benthamiana* leaves were collected for metabolite analysis after 4–5 days. The infiltrated *N. benthamiana* was lyophilized, ground, and weighed to 10 mg. The powder was extracted with 1 mL saponification solution (EtOH:H$_2$O:KOH 9:1:1 v:v:w) at 65 °C for 2 h. The extracts were then evaporated until dry and re-extracted with 500 µL ethyl acetate by shaking for 2 min. After an equal volume of H$_2$O was added, the samples were vortexed, and 100 µL of the upper organic phase was transferred to a fresh vial and dried. Trimethylsilylation was carried out with 50 µL of N-methyl-N-(trimethylsilyl)-trifluoroacetamide (Sigma–Aldrich, Darmstadt, Germany). The trimethylsilylation samples were analyzed using GC–MS.

### Functional characterization of *P450* genes in engineered yeast

Yeast strain Y0 was derived from strain Cenpk2-1D, which was purchased from EUROSCARF, by knocking out *BTS1* and *ERG27* genes using CRISPR/Cas9. Gene cassettes of Ppgk-tHMG-Tadh1, Ptef1-ERG1-Tpgk, Ptdh3-ERG20 + ERG9-TCYC1, and PpgK-β-AS-Tadh1 were amplified from the corresponding plasmids. Then, these cassettes and the URA3 marker gene were integrated into the delta site of strain Y0. The strain producing the highest **7** was designated strain Y1-20-6. The ORFs of *AcCYP716A275*, *BfCYP716Y1*, *AcCYP716A278*, and *GmCYP72A69* were subcloned and inserted into the yeast expression vector pESC-HIS, and the recombination constructs were cotransformed into yeast strain Y1-20-6. The metabolite extract was followed as above for GC–MS analysis.

### Yeast-feeding assay and microsomes for glucuronosylation of triterpenoid aglycones

The yeast expression vector pESC-HIS constructs with *AcCSL1* or an empty vector were transformed into yeast *S. cerevisiae* WAT11. The recombinant yeast strains were first grown in 20 mL SD-His medium containing 20 g/L glucose at 30 °C for 24 h. 200 µL of strain solution was used with urine glucose test strips to ensure that glucose was completely digested. After the glucose test, the yeast cells were harvested by centrifugation at 1000 × *g* for 5 min and washed twice with ddH$_2$O. SD-His medium with 20 g/L galactose was used to resuspend the strains. One milliliter of each strain was used for the yeast in vivo reaction. The substrate **1** was supplemented at 100 µM into the cultures. After 16 h, ethyl acetate was used to stop the reaction, and the supernatant was concentrated at low pressure after ultrasonic centrifugation until the ethyl acetate was completely volatilized. The pellet was dissolved in 200 µL methanol and stored at 4 °C for further UPLC–MS analysis. Microsomal proteins were isolated and dissolved in a protein storage buffer (20% [v/v] glycerol, 100 mM Tris–HCl [pH 7.5]). The enzymatic activity of AcCSL1 was assayed in 100 µL of reaction volume, which contained 35 mM Tris–HCl (pH 7.4), 1 mM dithiothreitol, ~30 µg of microsomal proteins, 100 µM **1**, and 1 mM UDPGA. The assays were incubated for 1 h at 30 °C. The reactions were stopped using 100 µL of methanol. Finally, the extracts were centrifuged for 10 min at 14,000 × *g* and analyzed by UPLC–MS.

### Enzyme assay of AcUGTs and AcBAHDs

Full-length cDNA sequences of the *UGT* and *BAHD* genes from *A. chinensis* were obtained using KOD Plus-Neo according to the kit instructions (TOYOBO, Shanghai, China). PCR products of *UGT* genes were purified with an EasyPure Quick Gel Extraction Kit (TransGen, Beijing, China), digested with the corresponding restriction enzyme, and ligated into similarly predigested pMAL-c2x vectors or pET28a (New England BioLabs, USA). The recombinant MBP- or His-fusion proteins were purified using maltose resin according to the manufacturer's instructions (New England Biolabs, USA). To analyze the activity of nineteen recombinant UGTs in vitro, 100 µl final reactions were adopted consisting of 10 mM DTT, 50 mM Tris–HCl (pH 7.0), 0.1 mM substrates, 1 mM UDP-glucose as a sugar donor, and recombinant proteins (5–10 µg). After incubation for 1 h at 37 °C, the reactions were stopped with 100 µL methanol. Products were analyzed by HPLC after centrifugation at 14,000 × *g* for 10 min and filtration through a 0.22 µm micropore filter. The $K_M$ and $V_{max}$ of AcUGTs were measured using 1 mM UDP-Glc as the sugar donor and **4** as the sugar acceptor; the substrate concentration was set in a 0–400 µM range as the acceptor in Tris–HCl buffer (pH 7.0). For the acetylation of candidate AcBAHDs toward **1** and **12**, 1 mM acetyl-CoA, 100 µM substrate, and 10 µg purified AcBAHD protein in 100 µL Tris–HCl solution (50 mM, pH 8.0) were incubated at 28 °C for 2 h. The reaction was terminated with an equal volume of methanol. Finally, the extracts were centrifuged for 10 min at 14,000 × *g*, and the catalytic products were analyzed by LC–MS.

### Enzyme assay of Ac4CLs and AcF6'Hs

All AcF6'H and Ac4CL genes were then inserted into the pET-32a (+) vector and transformed into the *E. coli* BL21(DE3) expression strain. After incubating the bacterial culture, protein expression was induced by 1.0 mM IPTG for 22 h at 16 °C. The bacterial sediment was collected and resuspended in Tris–HCl (pH 6.5 for AcF6'H proteins or pH 8.0 for Ac4CL proteins). Protein purification was performed using the His-tag Protein Purification Kit (Beyotime, Beijing, China).

For the functional screen and kinetic assay to measure Ac4CL enzyme activity, spectrophotometry was used to measure the changes in absorbance values at wavelengths of 324 and 357 nm. Each 100 µL reaction mixture was prepared with a moderate quantity of Ac4CL protein, 50 mM Tris–HCl (pH 8.0), 0.5 mM adenosine 5′-triphosphate, 0.5 mM MgCl$_2$, 0.03 mM CoA, and 2 mM substrate **13**. To determine the kinetic values of Ac4CL1 and Ac4CL2, Substrate **13** at different concentrations ranging from 0.01 to 0.48 mM was added to the reaction mixture. Enzyme kinetic values were calculated from the concentration of **14** generated per minute using a molar extinction coefficient of $\varepsilon = 18,000 \, M^{-1} \, cm^{-1}$.

To conduct an enzyme assay with AcF6'Hs, each 100 µL reaction mixture was prepared with a moderate quantity of protein, 100 mM Tris–HCl (pH 6.5), 1.0 mg/mL BSA, 0.005 mM FeSO$_4$, 1 mM disodium 2-oxoglutarate, 0.1 mM sodium ascorbate, and **14** at concentrations ranging from 0.005 to 0.3 mM to determine kinetic values and 2 mM to evaluate the candidate protein activity. The reaction was conducted at 30 °C for 2 min, 20 µL of 3 M NaOH was added, and the mixture was incubated at 37 °C for 10 min. Then, 20 µL of acetic acid was added, and the reaction mixture was centrifuged at 14,000 × *g* for 10 min. The supernatant was filtered through a 0.2 µm membrane filter and analyzed using LC–MS. Negative ion mode was used for product detection, and a C18 Waters Acquity UPLC® BEH C18 column (1.7 µm, 100 mm × 2.1 mm) was used for chromatography, with flow phase A consisting of 0.1% formic acid and water and flow phase B consisting of acetonitrile. The flow rate was set at 0.3 mL/min, and the column temperature was maintained at 40 °C. The gradient program of the mobile phase was as follows: 5% B at 0 min; 5% B at 2 min; 6% B at 13 min; 95% B at 13.1 min, 95% B at 14.1 min; and 5% B at 14.5 min. To determine the kinetic values, a fluorescence marker was used to detect the fluorescence values at 345 nm/462 nm using an enzyme-linked immunosorbent assay. The enzyme kinetic parameters were calculated using GraphPad Prism 8.0.2.

### LC–MS/MS and GC–MS/MS analysis for identification of enzymatic products

LC–QQQ–MS or LC–TOF–MS was used to analyze the catalytic products of AcUGTs, AcBAHDs, and AcF6'H. The LC–MS utilization and parameters are listed in Supplementary Table 16.

For the identification of β-amyrin and corresponding hydroxylated products, GC–QQQ–MS analyses were performed on an

Agilent Technology 7890 GC coupled with a 7000C Triple Quadrupole MS (Agilent Technologies, Santa Clara, CA, USA). One microliter of the sample was injected into splitless mode with an injector temperature of 250 °C. The column was DB-5 ms (Agilent Technologies, Santa Clara, CA, USA), 30 m × 0.25 mm i.d. × 0.25-μm film thickness, and was connected by a purged ultimate union (PUU) to provide sample separation. Helium was utilized as the carrier gas at a constant flow rate of 1.2 mL/min. The temperature was set as follows: start at 170 °C for 2 min, increase to 300 °C at 20 °C/min, and maintained at 300 °C for 11.5 min (the total run time was 20 min). The MS transfer line and ion source temperatures were set to 300 and 280 °C, respectively. Full mass spectra were generated by scanning within the $m/z$ range of 50–700 amu with a solvent delay of 8 min. **7**, **8**, and **9** were identified by comparing the retention time and mass spectra with the authentic standards **7**, and the catalytic reaction products of BfCYP716Y1 and GmCYP72A69, respectively.

### Purification and identification of 22-*O*-acetylprotoaescigenin

Bacterial cells expressing pET28a-AcBAHD6 recombinant proteins were harvested from 4 L of culture medium and resuspended in 480 mL of phosphate buffer. Crude enzymes were obtained by ultrasonication at 4 °C and then mixed with 50 mM **1** and 30 mM acetyl-CoA. A sufficient reaction was performed at 30 °C and 180 rpm for 5 h. Then, the reaction mixture was extracted three times with an equal volume of ethyl acetate, evaporated to dryness, and dissolved in 6 mL of ethyl acetate. Compound **11** was purified from this organic extract using preparative HPLC. The preparative HPLC was performed on an Agilent 1100 HPLC system equipped with a variable wavelength detector (VWD) and an evaporative light scattering detector (ELSD). The separation was performed on a 250 × 10 mm i.d. × 2.5 μm Cosmosil C18 column using water (A) and acetonitrile (B) at 2.5 mL/min with the following gradient: 0–55 min, 40% B; 56–60 min, 100% B; 61–70 min, 40% B. The injected volume was 100 μL, and the target product was collected between 21 and 23 min. The collected fractions were pooled and dried down through nitrogen blowing (to remove acetonitrile under a temperature of 0 °C) and freeze-dried (to remove water). This process yielded 1.5 mg of **11**, further subjected to NMR in dimethylsulfoxide-d6. NMR spectra were recorded on a Bruker Avance Neo 600 MHz magnetic resonance spectrometer, with tetramethylsilane (TMS) as an internal standard. Chemical shifts are reported in parts per million (ppm) relative to the residual solvent.

### De novo experiment of aesculin production

The derivative plasmids pZE12-luc (high copy number) and pCS27 (medium copy number) were used for pathway assembly. The plasmids pCS-TPTA-HpaBC and pCS-TAL were constructed in our previous study[72,73]. The RgTAL gene was amplified using pCS-TAL as the template. Plasmid pZE-649T was constructed by inserting *AcF6'H1*, *Ac4CL2*, *AcUGT92G7*, and *RgTAL* into pZE-12-luc using *Kpn* I, Sph I, *BamH* I, *Not* I, and *Xho* I. Information on the plasmids and the strains used is summarized in Supplementary Table 17. The primers used in this study are listed in Supplementary Table 18. For de novo production of **5**, 1 mL of overnight seed cultures was inoculated into 50 mL of M9Y media supplemented with appropriate antibiotics. The strain BW1 was cultivated at 37 °C and 200 rpm for 3 h and then induced with 0.5 mM IPTG. The induced cultures continued to grow at 30 °C and 200 rpm. Samples were obtained every 24 h. Cell growth was monitored by measuring the optical density at 600 nm, and the concentrations of the product and the intermediates were analyzed by HPLC.

### Reporting summary

Further information on research design is available in the Nature Portfolio Reporting Summary linked to this article.

## Data availability

The raw data of genome and transcriptome sequencing generated in this study have been deposited in the Genome Sequence Archive in BIG Data Center, Beijing Institute of Genomics (BIG), Chinese Academy of Sciences, under accession code CRA009101 and CRA009093. The assembled genome and gene structures of *A. chinensis* are available at Figshare [https://doi.org/10.6084/m9.figshare.21350865][93]. Source data are provided in this paper.

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

## Acknowledgements

This study was supported by scientific and technological innovation project of China Academy of Chinese Medical Sciences (CI2021A04112, CI2021A04117); the National Natural Science Foundation of China (82274037, 82304673, 21978015); National Key R&D Program of China (2019YFC1711100); Fundamental Research Funds for the Central public welfare research institutes (ZZ13-YQ-097); the Opening Project of State Key Laboratory of Tree Genetics and Breeding; the Key Laboratory of Plant Germplasm enhancement and Specialty Agriculture, Wuhan Botanical Garden, Chinese Academy of Sciences; and State Key Laboratory of Natural and Biomimetic Drugs. We also thank Create (Beijing) Technology Co., Ltd for supporting MALDI imaging.

## Author contributions

W.S., Z.C.X., and S.L.C. conceived and designed the research. Q.G.Y., R.R.G., H.H.W., X.H.X, Chong Xie, H.Y., Y.L.M., X.T.W., W.Q.C., and Chao Xiong performed the experiments. X.X.M. analyzed the results. H.Y. collected the materials. D.R.N named all the Aesculus genome P450 sequences. W.S., Z.C.X., X.X.S., Q.G.Y., and H.H.W. wrote the manuscript. Z.G.H., C.X.W., Ziyan Xie, Zheyong Xue, and P.S. reviewed and edited the manuscript. All authors have read and approved the manuscript for publication.

## Competing interests

The authors declare no competing interests.
