## [Peer Review File · Nature Communications]

Characterization of the horse chestnut genome reveals the evolution of aescin and aesculin biosynthesisReviewers' Comments:

Reviewer #1:

Remarks to the Author:

Report on "Horse chestnut tree genome reveals the evolutionary mechanism of aescin and aesculin biosynthesis" by Sun et al. In this study, to uncover the biosynthesis of medicinal molecular aescin and aesculin, the authors sequenced and assembled a genome of horse chestnut, and characterized a whole-genome duplication event. They found two gene clusters that are involved in aescin biosynthesis. Five candidate genes involved in aescin biosynthesis and two UGTs involved in aesculin formation were functionally identified. They also analyzed the organization and evolution of clusters across angiosperms through comparative genomics approach.

This job as a whole provides a good example for understanding the evolution of gene clusters related to the medicinal metabolites. However, the approach of utilizing multi-omics to discover the gene clusters and identify the candidate genes is generalized in the field of plant natural products. Importantly, although candidate genes including one OSC, two CYPs and one CSL involved in aescin biosynthesis in *A. chinensis* were functionally identified, the same functional enzymes have been already reported, as shown in the manuscript, authors used them as positive controls for candidate identification. In addition, some data is not clear, even not consistent with the descriptions in manuscript. The logic in the manuscript is not completely clear, particularly in the section of coumarin (aesculin) biosynthesis, in which only UGTs was functionally identified. And many other genes involved in aesculin biosynthesis in *A. chinensis* were not tested even if authors discovered them in *A. chinensis* genome. It is suggested to well organize the manuscript such that easy to follow. Therefore, I don't recommend this paper for publication. Below is my suggestions and comments, hope them useful for manuscript improvement.

1. Line 98: It is possibly not suitable to describe the metabolite distribution using triterpenoid and coumarin here. Because aescin or aesculin (or fraxin) is just one of triterpenoid or coumarin.
2. Line 99 to 103: Authors performed MALDI-MSI analysis to show the metabolite distribution in fruit structures, including fraxin, aesculioside A, aesculioside G, escin V and isoescin IIb. Is there any connection to aescin and aesculin biosynthesis? Why did author measure these compounds if they focus on aescin and aesculin biosynthesis? It is not clear in the manuscript.
3. Line 104 and 108: Authors mentioned stems samples, but there is no any stems data in the figures. Please explain why?
4. In figure 1D: Fratin in legend should be Fraxin?
5. In figure 1D: Why did author measure Fraxetin, Esculetin, coumarins if they focus on aescin and aesculin biosynthesis? It is not clear in the manuscript. Which compounds were included in the total coumarins? They are not clear in the legend.
6. In figure 1E: Why did author measure Sarcocarp and Epicarp? Is there any connection to aescin and aesculin biosynthesis?
7. In figure 1E: What's the total triterpenoids? All the triterpenoid compound in samples?
8. Based on introduction, authors aim to uncover the biosynthesis of escin Ia, escin Ib (referred to as aescin) and aesculin. However, in the result of Line 96 to 113, authors measure many other compounds besides aescin and aesculin. Authors don't clearly describe their relationship with aescin or aesculin. It is suggested to rearrange and simplify this part of result to avoid confuse.
9. In the first section of result, authors measured the metabolites in different tissues and different developmental stages of these tissues. However, in the next RNA-seq analysis, author only provided the data of different tissues. It seems useless for the measurement of metabolites in different developmental stages to the identification of the aescin candidate genes. Please provide the explanation why add this data in Figure 1E in the manuscript.
10. Line 221: It would be helpful to the general reader if add explanation why CYP716 family was targeted (perhaps along with the relevant reactions by CYP716 in aescin biosynthesis).
11. Line 224 to 227: Authors describe the AcCluster I with two OSCs, six CYPs, and five BAHDs. However, seven CYPs including four CYP716A and three CYP16BX, and five BAHDs were shown in Figure 3A, and seven CYPs and four BAHDs were shown in Figure S9. Please provide the explanation

for these differences.

12. It would be helpful if add the name of other candidate gene in Figure 3A, like AcCYP16A278, AcOSC6, and AcBAHD3.
13. Please provide the legends of other arrows indicating shown in Figure 3A.
14. Line 228 to 231: Authors described one OSC (AcOSC6), two CYPs (AcCYP716A278, AcCYP716BX1) and three BADH (AcBAHD1, AcBAHD3, AcBAHD5) showed co-expression in seed. However, in Figure 3B, these genes showed higher expression in Leaf, not agreement with high aescin level in the seeds. This data is not consistent with the description provided in manuscript. In addition, there is no any annotation for AcCYP716BX1, AcBAHD1, and AcBAHD5.
15. It would be helpful if add the name of all candidate genes in Figure 3B, particular for seed-specific expressed genes.
16. In Figure 3B, based on the legend of heat-map, the deep blue indicates higher gene expression, and deep red indicates lower gene expression. It seems that the legend is not consistent with the data shown in the heat-map. If heat-map data is right, please note that AcBAHD3 did not co-expressed with other candidate genes based on heat-map data.
17. In Figure 3B, Author provided the RNA-seq data of branch. However, In Figure 1, no metabolite measurement in branch was observed. How about aescin content in branch? Please provide the relevant data.
18. Line 241 to 244: Author mentioned that they performed WGCNA to screen glycosyltransferases and identified nine putative candidates. Please provide the data for these genes.
19. In the manuscript, authors found five gene clusters involved in triterpenoid biosynthesis, but only AcCluster I and II were analyzed through RNA-seq data. Any other seed-specific expressed genes exist in other three clusters? Please provide the relevant data.
20. GmCYP716Y1 in Line 264 don't match to that shown in Figure 3F.
21. Please provide the mass spectrum data of AcCSL1 and GmCSyGT1 activity in tobacco experiments.
22. For activity experiments of AcOSC6, AcCYP716A278, AcCYP716A275, AcCSL, authors used beta-amyrin standard, GmCYP72A69, GmCYP716Y1, GmCSL as positive control respectively. However, there is no positive control for AcBAHD3 activity experiments, and even no mass spectrum data. As such, NMR data should be provided to confirm the acylation position and product structure.
23. It is suggested to provide the result of AcBAHD1 and AcBAHD5 activity assays even if they have no activity to protoaescigenin. They can be considered as negative controls.
24. In Figure 3H legend, authors described they used E. coli expressing AcBAHD1 for activity experiments, yet in the manuscript and figure AcBAHD3 was shown. In addition, as shown in Figure 3B, AcBAHD3 seems no co-expression with other candidates. Please carefully check them.
25. It is suggested to provide SDS-PAGE verification result of purified AcBAHD1, AcBAHD3, AcBAHD5 when their functional activity was identified in vitro.
26. In Figure 3H, what is the CK? Please clearly indicate it in legend.
27. Line 325 to 327: "Syntenic analysis....." Should be a figure or reference cited here?
28. For the identification of AcUGTs functional activity, did in vitro assays were carried out? If so, please provide SDS-PAGE verification result of purified AcUGTs from E. coli.
29. Line 380 to 387: Authors describe that seven PALs, three C4Hs, five C3Hs, four 4CLs, five F6Hs, three COMTs, four CcoAOMTs, and five S8Hs involved in aesculin and fraxin biosynthesis were identified from the A. chinensis genome. But these genes showed different expression patterns in tissues as shown in Figure S19A. Why did authors not screen aesculin or fraxin candidate genes through RNA-seq data by co-expression approach? Why did authors not identify the functional activities of above candidate genes and uncover the aesculin biosynthesis in A. chinensis?
30. Figure 5A shows the result of esculetin utilization, while Figure 5B shows the result of fraxetin utilization, which are not consistent with the description in manuscript.
31. Figure S19B and D is the repeat of that in Figure 5A and B. It is suggested to delete one.
32. Please indicate what is the CK in Figure 5A and B legend.
33. Please provide the MS spectrums of aesculin generated from UGT92G7+esculetin, and UGT84A56+esculetin. Similarity, MS spectrums of fraxin generated from UGT92G7+fraxetin, and UGT71A47+fraxetin also need to be provided.
34. Although only three UGTs (UGT92G7, UGT84A56, UGT71A47) were identified with functional

activity to esculetin or fraxetin, other UGTs activity result should be shown even if no activity.

35. Why the MS data of aesculin in Figure S19C is different from that of aesculin in Figure S20?

36. In section of De novo production of aesculin in *E. coli*: Why did authors use the heterologous genes, such as F6H and 4CL from *A. thaliana*, or F6P and 4CL1 from *R. glutinis*, not use the native genes in *A. chinensis* to establish the aesculin pathway in *E. coli*? In the aesculin pathway in *E. coli*, it seems only UGT84A56 is from *A. chinensis*. It should use the candidate genes from *A. chinensis* to construct the aesculin pathway such that confirm their functional activities.

Reviewer #2:

Remarks to the Author:

The manuscript by Sun et al presents a massive amount of work in characterizing the horse chestnut tree genome and the evolution of Aescin and Aesculin biosynthesis. The work encompasses a chromosome scale genome assembly, phylogenetics, metabolomics and cloning and recombinant expression of a vast number of enzymes. The study is extensive, important and provided very clear results that greatly enhance our understanding of the evolution of these important phytochemicals. The work is clearly well planned and executed - the figures are exemplary and the authors should be congratulated on this.

I have no concerns regarding either the experimental work or the interpretation of the results. However, whilst understandable the article needs to be improved in terms of language and/syntax. I suggest that this needs a further deep professional edit. However, I add a (non-comprehensive) list of things I spotted

- 1) the word mechanism in the title makes little sense I suggestreveals insights into the evolution of...instead.
- 2)line 81 "This unknown limit the efforts" is not very elegant and should be rewritten
- 3) D- and L- forms of chemicals, the D- and L- should be set in one font size smaller following the chemical nomenclature.
- 4) line 422 I guess you mean chorismate, as far as I know chorismite is a rock!
- 5) I was not fully happy with the conclusion I think this should better be a contextualization of the main results within the scientific literature rather than a mere summary of the main findings. In this way a future perspective for the work could also be added.

Reviewer #3:

Remarks to the Author:

This is a very comprehensive paper that covers bioinformatics, evolution, spatial metabolomics and biochemistry. Its impressive the amount of work that has been put in to the paper and the supplementary data. The data will be an example of how elucidation of complex plant specialized metabolites in a tree. The paper is very well written and I only have minor comments, mainly regarding the biochemistry.

P450s are listed as CYP450, CYP and P450, I suggest using the term P450.

It is mentioned that K_m and V_{max} have been calculated, but I have not been able to see where the data is. I would like to encourage that authors to engage into a discussion of the impact on K_m and if the data represent in planta relevant activities, or that eg the K_m is so high that it may not be a realistic reactions measured. K_m s have been published for other UGTs in triterpenoid biosynthesis as well as other plant specialized metabolites.

One of the OSC in figure S13 has an unusual long branch length which indicates that the annotation is

incorrect.

In the discussion of especially CYP72A substrate specificities there are other examples in the literature of where CYP72A may hydroxylate, adding to the discussion of convergent versus divergent evolution of the diverse CYP72A activities.

Reviewer #1 (Remarks to the Author):

Report on “Horse chestnut tree genome reveals the evolutionary mechanism of aescin and aesculin biosynthesis” by Sun et al. In this study, to uncover the biosynthesis of medicinal molecular aescin and aesculin, the authors sequenced and assembled a genome of horse chestnut, and characterized a whole-genome duplication event. They found two gene clusters that are involved in aescin biosynthesis. Five candidate genes involved in aescin biosynthesis and two UGTs involved in aesculin formation were functionally identified. They also analyzed the organization and evolution of clusters across angiosperms through comparative genomics approach.

This job as a whole provides a good example for understanding the evolution of gene clusters related to the medicinal metabolites. However, the approach of utilizing multi-omics to discover the gene clusters and identify the candidate genes is generalized in the field of plant natural products. Importantly, although candidate genes including one OSC, two CYPs and one CSL involved in aescin biosynthesis in *A. chinensis* were functionally identified, the same functional enzymes have been already reported, as shown in the manuscript, authors used them as positive controls for candidate identification. In addition, some data is not clear, even not consistent with the descriptions in manuscript. The logic in the manuscript is not completely clear, particularly in the section of coumarin (aesculin) biosynthesis, in which only UGTs was functionally identified. And many other genes involved in aesculin biosynthesis in *A. chinensis* were not tested even if authors discovered them in *A. chinensis* genome. It is suggested to well organize the manuscript such that easy to follow. Therefore, I don't recommend this paper for publication. Below is my suggestions and comments, hope them useful for manuscript improvement.

Response: Thank you for your constructive comments. We appreciate the time and effort you have dedicated to providing valuable feedback on our manuscript. We apologize for the unclear data and inconsistent descriptions presented in the previous version. Based on your advice, we have provided *in vitro* functional identification of the novel AcBADH6 enzyme, which is related to the acetylation of C22-O-protoaescigenin and desacetylaescin I in the triterpenoid biosynthesis pathway. We have also conducted biochemical investigations on Ac4CLs, AcF6Hs, and AcUGTs in aesculin biosynthesis, as well as re-*de novo* production of aesculin using them in *E. coli*. Additionally, we have carefully revised our manuscript and incorporated most of your suggestions. We hope that our revised manuscript will be reconsidered for publication.

1. Line 98: It is possibly not suitable to describe the metabolite distribution using triterpenoid and coumarin here. Because aescin or aesculin (or fraxin) is just one of triterpenoid or coumarin.

Response: We have revised this section in response to the reviewers' feedback. To improve clarity, we have replaced vague terms such as triterpenoid and coumarin with more specific descriptions of the aescin or aesculin ingredients that are directly relevant to this manuscript. Additionally, after careful consideration, we have decided to remove content related to fraxin as it is not a medicinal compound currently used in clinical practice. We believe these changes will enhance the clarity of our manuscript.

2. Line 99 to 103: Authors performed MALDI-MSI analysis to show the metabolite distribution in fruit structures, including fraxin, aesculiside A, aesculoside G, escin V and isoescin IIb. Is there

any connection to aescin and aesculin biosynthesis? Why did author measure these compounds if they focus on aescin and aesculin biosynthesis? It is not clear in the manuscript.

Response: We acknowledge the reviewer's suggestion that our description was insufficiently clear. Accordingly, we have revised our focus to specifically address the coumarin compounds (esculetin and aesculin) and triterpenoid compounds (protoescigenin and escin Ia or escin Ib) relevant to the topic of our manuscript. However, we regret to report that we were only able to detect escin Ia or escin Ib in the fruit sample using MALDI-MS.

3. Line 104 and 108: Authors mentioned stems samples, but there is no any stems data in the figures. Please explain why?

Response: Thank you for pointing this out. We apologize for our mistake in mixing up the terms "stem" and "branch" and for omitting some data. After carefully reviewing the entire text, we have confirmed that "branch" is the correct term for organ description, and have provided RNA-seq and metabolite detection data for this organ.

4. In figure 1D: Fratin in legend should be Fraxin?

Response: Thank you for your suggestions. We appreciate your comments and have taken note of the mistake. The addition of the component fraxin has made the main point of the article less clear. In response, we have decided to remove the discussion of fraxin and its precursor fraxetin from our manuscript, as they are not clinically used as medicine.

5. In figure 1D: Why did author measure Fraxetin, Esculetin, coumarins if they focus on aescin and aesculin biosynthesis? It is not clear in the manuscript. Which compounds were included in the total coumarins? They are not clear in the legend.

Response: Thank you for your suggestions. We acknowledge that fraxetin and esculetin are indeed precursors of fraxin and aesculin, respectively. However, as fraxin and its precursor fraxetin are not clinically relevant compounds, we have decided to remove the results and discussion of these compounds from our manuscript. Instead, we have heeded your recommendation to shift our focus towards the biosynthesis of escin Ia, including its precursor protoescigenin and isomer escin Ib, as well as aesculin and its precursors esculetin and aesculin, which are more directly relevant to the theme of our manuscript. We also apologize for the incorrect inclusion of total coumarin and triterpenoid content, which resulted from a misconception of the compounds. We have removed this information from our manuscript.

6. In figure 1E: Why did author measure Sarcocarp and Epicarp? Is there any connection to aescin and aesculin biosynthesis?

Response: To narrow down the candidate genes associated with aescin and aesculin biosynthesis, we aimed to identify differential distribution of aescin and aesculin related compounds among various tissues. As fruits are a major resource for extracting aescin, we systematically observed the distribution in the sarcocarp, epicarp, and seed. However, due to a lack of RNA-Seq data for the sarcocarp, we have removed the data and description of aescin and aesculin accumulation in the sarcocarp from our manuscript to improve clarity.

7. In figure 1E: What's the total triterpenoids? All the triterpenoid compound in samples?

Response: Thank the reviewer for your comments. The detected escin Ia, escin Ib, and protoescigenin in figure 1E were defined as total triterpenoids. We acknowledged that this wording was problematic, so we have removed it to reflect only accurate quantification of protoescigenin, escin Ia and escin Ib. We have re-drawn the Figure 1.

8. Based on introduction, authors aim to uncover the biosynthesis of escin Ia, escin Ib (referred to as aescin) and aesculin. However, in the result of Line 96 to 113, authors measure many other compounds besides aescin and aesculin. Authors don't clearly describe their relationship with aescin or aesculin. It is suggested to rearrange and simplify this part of result to avoid confuse.

Response: Thank you for your suggestions. We agree that the detection of some compounds is not directly related to the aescin and aesculin that we have focused on. Therefore, we have reorganized the relevant content and simplified the description of this part. In total, we detected five compounds including protoaescigenin, escin Ia, escin Ib, esculetin, and aesculin.

9. In the first section of result, authors measured the metabolites in different tissues and different developmental stages of these tissues. However, in the next RNA-seq analysis, author only provided the data of different tissues. It seems useless for the measurement of metabolites in different developmental stages to the identification of the aescin candidate genes. Please provide the explanation why add this data in Figure 1E in the manuscript.

Response: We thank the reviewer for their suggestions. Our initial selection of tissues and organs from different harvest seasons allowed us to identify clear differences in metabolite levels, which helped us to design the subsequent RNA-seq experiment. However, we did not perform RNA-seq on tissues from different seasons. As the reviewers noted, we did not specifically describe this part of the data in the full text, so we decided to remove it to simplify the overall results. In addition, we have chosen the same tissues for the data presentation of metabolite content and RNA-seq.

10. Line 221: It would be helpful to the general reader if add explanation why CYP716 family was targeted (perhaps along with the relevant reactions by CYP716 in aescin biosynthesis).

Response: Thanks to the reviewer for your suggestions. We have revised this sentences as follows. "To find BGCs involved in triterpenoid biosynthesis in the *A. chinensis* genome, we searched for genomic region containing OSCs and /or tailoring enzymes (e. g. the most common CYP716 family that modifies triterpene skeletons) known to act in such metabolism. This led to discovery of four BGCs containing OSCs and one BGC containing CYP716 genes, known as AcClusters I-V, which are then implicated in triterpenoid biosynthesis in the *A. chinensis* genome (Figure S9)."

11. Line 224 to 227: Authors describe the AcCluster I with two OSCs, six CYPs, and five BAHDs. However, seven CYPs including four CYP716A and three CYP16BX, and five BAHDs were shown in Figure 3A, and seven CYPs and four BAHDs were shown in Figure S9. Please provide the explanation for these differences.

Response: We apologize for the mistake in our previous version. AcCluster I actually includes two OSCs, seven CYPs (four CYP716A and three CYP716BX), and five BAHDs. We have made the necessary revisions to both the main text and Figure S9.

12. It would be helpful if add the name of other candidate gene in Figure 3A, like AcCYP16A278, AcOSC6, and AcBAHD3.

Response: Following your comment, we revised the Figure 3A.

13. Please provide the legends of other arrows indicating shown in Figure 3A.

Response: We have revised the legends of Figure 3A. The grey arrows signify the non-syntenic genes.

14. Line 228 to 231: Authors described one OSC (AcOSC6), two CYPs (AcCYP716A278, AcCYP716BX1) and three BADH (AcBAHD1, AcBAHD3, AcBAHD5) showed co-expression in seed. However, in Figure 3B, these genes showed higher expression in Leaf, not agreement with high aescin level in the seeds. This data is not consistent with the description provided in manuscript. In addition, there is no any annotation for AcCYP716BX1, AcBAHD1, and AcBAHD5.

Response: We are sorry for the mistake. The color key for gene expression value in Figure 3B is wrongly presented, and we have revised it. Following your comments, we added the annotation for all candidate genes in the Figure 3B.

15. It would be helpful if add the name of all candidate genes in Figure 3B, particular for seed-specific expressed genes.

Response: Following your comment, we have added the name of all candidate genes and revised Figure 3A and B.

16. In Figure 3B, based on the legend of heat-map, the deep blue indicates higher gene expression, and deep red indicates lower gene expression. It seems that the legend is not consistent with the data shown in the heat-map. If heat-map data is right, please note that AcBAHD3 did not co-expressed with other candidate genes based on heat-map data.

Response: We are sorry for the mistake. The color key for gene expression value in Figure 3B is wrongly presented, and we have revised it.

17. In Figure 3B, Author provided the RNA-seq data of branch. However, In Figure 1, no metabolite measurement in branch was observed. How about aescin content in branch? Please provide the relevant data.

Response: Thanks for your advice. We have added the LC-MS/MS analysis for the branch samples in Figure 1. Now, the tissues for RNA-Seq and metabolite detection are consistent.

18. Line 241 to 244: Author mentioned that they performed WGCNA to screen glycosyltransferases and identified nine putative candidates. Please provide the data for these genes.

Response: Thanks for your comment. We supplied the gene list and annotation of seed specific “turquoise” module from WGCNA co-expression analysis in Table S16. In addition, the potential OSC, CSLs, UGTs, CYP450s, and BAHDs have also been marked.

19. In the manuscript, authors found five gene clusters involved in triterpenoid biosynthesis, but only AcCluster I and II were analyzed through RNA-seq data. Any other seed-specific expressed genes exist in other three clusters? Please provide the relevant data.

Response: Following your advice, we have added the gene expression value of other three AcClusters in Table S15. The gene expression profile showed that there are not any seed-specific expressed gene distributed in these three clusters.

20. GmCYP716Y1 in Line 264 don't match to that shown in Figure 3F.

Response: Sorry for the mistake. We have made a revision to the main text and Figure 3E, changing GmCYP716Y1 to BfCYP716Y1.

21. Please provide the mass spectrum data of AcCSL1 and GmCSyGT1 activity in tobacco experiments.

Response: I am sorry to puzzle you for wrong description. We use the yeast to verify the function of AcCSL1 and GmCSyGT1, rather tobacco. The description has been changed to “recombinant expression in yeast”. And the MS spectrums were shown in Figure S16.

22. For activity experiments of AcOSC6, AcCYP716A278, AcCYP716A275, AcCSL, authors used beta-amyrin standard, GmCYP72A69, GmCYP716Y1, GmCSL as positive control respectively. However, there is no positive control for AcBAHD3 activity experiments, and even no mass spectrum data. As such, NMR data should be provided to confirm the acylation position and product structure.

Response: We re-evaluated the enzymatic function of AcBAHD1, 3, and 5, along with the newly discovered highly expressed AcBAHD6 in seeds, towards protoaescigenin. Our findings revealed that AcBAHD6 demonstrated superior catalytic function at the 22-O-acylation position of protoaescigenin, and this result was confirmed by NMR data and LC-QTOF-MS. Despite the challenges associated with the purification process of 22-O-acetylprotoaescigenin, we remained persistent for nearly four months and successfully enriched the compound. We would like to express our gratitude to the reviewer for their patience. Regrettably, the catalytic function of AcBAHD3 towards protoaescigenin was found to be very low. Additionally, we discovered that AcBAHD3 and AcBAHD6 could use acetyl-CoA as an acetyl donor to catalyze desacetylaescin I, resulting in the formation of escin Ia (C22-O-acetylation of desacetylaescin I). AcBAHD6 exhibited a superior C22-O-acetylation function towards desacetylaescin I when compared to AcBAHD3.

23. It is suggested to provide the result of AcBAHD1 and AcBAHD5 activity assays even if they have no activity to protoaescigenin. They can be considered as negative controls.

Response: According to your kindly suggestion, the negative results of AcBAHD1/5 were presented in the Figure S18.

24. In Figure 3H legend, authors described they used E. coli expressing AcBAHD1 for activity experiments, yet in the manuscript and figure AcBAHD3 was shown. In addition, as shown in Figure 3B, AcBAHD3 seems no co-expression with other candidates. Please carefully check them.

Response: We apologize for the unclear description. Initially, we screened AcBAHD1, AcBAHD3, and AcBAHD5 in AcCluster I for functional verification, as they exhibit higher expression in the seed. However, only AcBAHD3 demonstrated acylation activity with low capacity towards protoaescigenin. Due to the low catalytic activity of AcBAHD3, we proceeded to select AcBAHD6, which is expressed specifically in the seed based on the WGCNA network and is not distributed in

AcClusters. We have revised Figure 3B, H, and I to demonstrate the expression and catalytic activity of the candidate AcBAHDs.

25. It is suggested to provide SDS-PAGE verification result of purified AcBAHD1, AcBAHD3, AcBAHD5 when their functional activity was identified *in vitro*.

Response: We have added the SDS-PAGE verification results of recombinant AcBAHD1, AcBAHD3, AcBAHD5, and AcBAHD6 in Figure S17.

26. In Figure 3H, what is the CK? Please clearly indicate it in legend.

Response: CK means negative control, that is, the reaction sample lacking recombinant proteins, we have revised the Figure 3H and 3I to present the functional verification of AcBAHD3 and AcBAHD6. We have presented the negative control in Figure S18 in detail.

27. Line 325 to 327: “Syntenic analysis.....” Should be a figure or reference cited here?

Response: Following your advice, we have cited the Figure 5A.

28. For the identification of AcUGTs functional activity, did *in vitro* assays were carried out? If so, please provide SDS-PAGE verification result of purified AcUGTs from *E. coli*.

Response: The SDS-PAGE gel picture with purified recombinant AcUGT proteins was inserted in Figure S26.

29. Line 380 to 387: Authors describe that seven PALs, three C4Hs, five C3Hs, four 4CLs, five F6Hs, three COMTs, four CcoAOMTs, and five S8Hs involved in aesculin and fraxin biosynthesis were identified from the *A. chinensis* genome. But these genes showed different expression patterns in tissues as shown in Figure S19A. Why did authors not screen aesculin or fraxin candidate genes through RNA-seq data by co-expression approach? Why did authors not identify the functional activities of above candidate genes and uncover the aesculin biosynthesis in *A. chinensis*?

Response: Thank you for your suggestion. Due to the lack of clear and specific distribution of aesculin and esculetin compounds, it was challenging to screen relevant genes using the co-expression model. Following the reviewer's advice, we have included gene functional identification and enzyme activity assays for Ac4CLs and AcF6'Hs in this study. Additionally, we have reconstructed the engineering strain of the medicinal ingredient aesculin by combining Ac4CL2 and AcF6'H1 with the previously discovered AcUGT92G7.

30. Figure 5A shows the result of esculetin utilization, while Figure 5B shows the result of fraxetin utilization, which are not consistent with the description in manuscript.

Response: According to your suggestion, in order to make the entire text more fluent and clear, we have decided to remove the content related to fraxin compound synthesis. We have deleted the catalyzation of UGT71A47 and UGT92G7 towards to fraxetin, and revised the Figure 4. AcUGT84A56 and AcUGT92G7 were found to utilize esculetin (Figure 4A, D).

31. Figure S19B and D is the repeat of that in Figure 5A and B. It is suggested to delete one.

Response: We have revised original Figure 5 into Figure 4, and the corresponding supplemental Figures.

32. Please indicate what is the CK in Figure 5A and B legend.

Response: CK means reaction sample lacking recombinant proteins, we have added the description in the legend of new Figure 4C and D.

33. Please provide the MS spectrums of aesculin generated from UGT92G7+esculetin, and UGT84A56+esculetin. Similarity, MS spectrums of fraxetin generated from UGT92G7+fraxetin, and UGT71A47+fraxetin also need to be provided.

Response: We have added the MS spectrum data of the aesculin produced by AcUGT92G7 and AcUGT84A56 in Figure S27. In addition, we have deleted the catalyzation of AcUGT71A47 and AcUGT92G7 towards fraxetin.

34. Although only three UGTs (UGT92G7, UGT84A56, UGT71A47) were identified with functional activity to esculetin or fraxetin, other UGTs activity result should be shown even if no activity.

Response: According to your advice, we have added the negative results in the Figure S26.

35. Why the MS data of aesculin in Figure S19C is different from that of aesculin in Figure S20?

Response: In the previous version, the validation of UGT functional genes catalyzing esculetin and the *de novo* aesculin experiment were conducted in two separate laboratories, resulting in slight differences in the mass spectrometry data for aesculin. However, the MS results for aesculin were based on standard data. We re-designed the *de novo* production of aesculin in *E. coli*, and then the product was identified using the same LC-MS/MS platform. Therefore, we did not present the MS spectrum of aesculin again in the section of “*de novo* production of aesculin in *E. coli*”.

36. In section of De novo production of aesculin in *E. coli*: Why did authors use the heterologous genes, such as F6H and 4CL from *A. thaliana*, or F6P and 4CL1 from *R. glutinis*, not use the native genes in *A. chinensis* to establish the aesculin pathway in *E. coli*? In the aesculin pathway in *E. coli*, it seems only UGT84A56 is from *A. chinensis*. It should use the candidate genes from *A. chinensis* to construct the aesculin pathway such that confirm their functional activities.

Response: Following your advice, we functionally identified the catalytic activities of Ac4CLs, AcF6’Hs, and AcUGTs from *A. chinensis* for aesculin biosynthesis. The results showed that Ac4CL2, AcF6’H1, and AcUGT92G7 presented superior catalytic properties than other candidate enzymes. Therefore, the characterized enzymes Ac4CL2, AcF6’H1 and AcUGT92G7 were used to assemble a *de novo* biosynthetic pathway for aesculin. The results showed that the final aesculin production at 72h reached 16.3±0.7 mg/L.

Reviewer #2 (Remarks to the Author):

The manuscript by Sun et al presents a massive amount of work in characterizing the horse chestnut tree genome and the evolution of Aescin and Aesculin biosynthesis. The work encompasses a chromosome scale genome assembly, phylogenetics, metabolomics and cloning and recombinant expression of a vast number of enzymes. The study is extensive, important and provided very clear results that greatly enhance our understanding of the evolution of these important phytochemicals.

The work is clearly well planned and executed - the figures are exemplary and the authors should be congratulated on this.

Response: We really appreciated your positive comments.

I have no concerns regarding either the experimental work or the interpretation of the results. However, whilst understandable the article needs to be improved in terms of language and/syntax. I suggest that this needs a further deep professional edit. However, I add a (non-comprehensive) list of things I spotted

1) the word mechanism in the title makes little sense I suggestreveals insights into the evolution of...instead.

Response: Thanks for your suggestion. We have revised the title name according to your suggestion.

2) line 81 "This unknown limit the efforts" is not very elegant and should be rewritten

Response: Thanks for your suggestion. We have rewritten this sentence.

3) D- and L- forms of chemicals, the D- and L- should be set in one font size smaller following the chemical nomenclature.

Response: We appreciated your suggestion. These words have been rewritten, and the entire paragraph has been checked for accuracy.

4) line 422 I guess you mean chorismate, as far as I know chorismite is a rock!

Response: We appreciated your suggestion. These words have been rewritten, and the entire paragraph has been checked for accuracy.

5) I was not fully happy with the conclusion I think this should better be a contextualization of the main results within the scientific literature rather than a mere summary of the main findings. In this way a future perspective for the work could also be added.

Response: We appreciated your suggestion. We have revised the conclusion following your advice.

“While the principles of evolution of metabolic pathways with multiple steps are less clear in plant, it is rather well understood individual enzymes have evolved through gene duplications to acquire metabolic diversity (79–81). The perspective to comprehend the underlying causes of compound origin is provided by the plant genes linked in the genomic region known as the BGC, which includes various types of encoding enzymes (82). Currently, much work takes advantage of the rich resource of genomic sequence to uncover complex evolutionary procedures that impact on the conservation and flexibility of BGC, which determine the appearance, maintenance, and innovation of chemical components in the scope of interspecies or intraspecies (14, 83–85). In this study, our genomic-driven tactics provided fresh perspectives on how synthesis and evolution of barrigenol-type triterpenoids across angiosperms. The evolutionary trajectories of BGCs among angiosperms revealed that the functional BGC leading to BAT biosynthesis was specifically assembled in Hippocastanoideae species, and dynamic evolution of species-specific via tandem duplication and WGD after the speciation of *A. chinensis*. Furthermore, without wild *Aesculus*, it is currently difficult to economically synthesize aescin and aesculin. Despite the fact that synthetic

biology offers a promising method for producing desired components in heterologous systems, enzymatic reactions responsible for their biosynthesis are still unexploited. The exquisite genome editing techniques might be used to bioengineer crop and microbial species to manipulate the multi-enzymes involved in the aescin and aesculin biosynthesis processes, allowing for the large-scale manufacture of high-value pharmaceutical components.”

Reviewer #3 (Remarks to the Author):

This is a very comprehensive paper that covers bioinformatics, evolution, spatial metabolomics and biochemistry. Its impressive the amount of work that has been put in to the paper and the supplementary data. The data will be an example of how elucidation of complex plant specialized metabolites in a tree. The paper is very well written and I only have minor comments, mainly regarding the biochemistry.

Response: We really appreciated your positive comments.

1. P450s are listed as CYP450, CYP and P450, I suggest using the term P450.

Response: I appreciate your suggestion. We use uniformly term P450 representing CYP450, CYP and P450.

2. It is mentioned that K_m and V_{max} have been calculated, but I have not been able to see where the data is. I would like to encourage that authors to engage into a discussion of the impact on K_m and if the data represent in planta relevant activities, or that eg. the K_m is so high that it may not be a realistic reactions measured. K_m s have been published for other UGTs in triterpenoid biosynthesis as well as other plant specialized metabolites.

Response: Sorry for missing this important kinetic data of AcUGT92G7 and AcUGT84A56 for esculetin. We have added its descriptions in the section of result and discussion. “The enzymatic products of AcUGTs with esculetin as the substrates were confirmed to be aesculin by comparing mass spectra and retention time to that of authentic standards (Figure S27). Enzymatic kinetic test showed that the affinity of AcUGT92G7 ($K_M=48.96 \mu M$) stronger than AcUGT84A56 ($177.25 \mu M$) for esculetin (Table S18).”.

3. One of the OSC in figure S13 has an unusual long branch length which indicates that the annotation is incorrect.

Response: Thanks for your comment. We have reconstructed the OSC phylogenetic tree. Please see the Figure S13.

4. In the discussion of especially CYP72A substrate specificities there are other examples in the literature of where CYP72A may hydroxylate, adding to the discussion of convergent versus divergent evolution of the diverse CYP72A activities.

Response: Following your advice, we described the hydroxylation activities of reported CYP716A and CYP72A members towards triterpenoids, and further discussed the independent evolution of CYP72A and CYP716A, which contributes to the C-21 oxidation of oleanane-type triterpenoids.

Reviewers' Comments:

Reviewer #1:

Remarks to the Author:

In the revised manuscript, my comments were satisfactorily answered. The current revision is well organized and clearer to follow. Regarding the novelty in this work, it still did not meet my expectations. But given the massive amount of work and well-revised manuscript, it should be encouraged. I recommend this report for publication in Nature Communications.

Reviewer #2:

Remarks to the Author:

I am fully satisfied with the revisions made by the authors. I have no further suggestions.

Reviewer #3:

Remarks to the Author:

Regarding figure 1. Use both trivial names and show structures to combine figures 1 A, B, and C so its easier for the reader to follow. The notation that Escin is "in the seeds" is rather unclear. Could the authors be more clear about which tissues in the seed?

Two CYP 72As are mentioned in the text; however there are more, eg CYP72A555 is involved in biosynthesis of saponins and CYP72s are involved in hydroxylation of gibberellins – suggesting that they have been recruited from that pathway. CYP72A enzymes catalyse 13-hydroxylation of gibberellins - PubMed (nih.gov) and The cytochrome P450 CYP72A552 is key to production of hederagenin-based saponins that mediate plant defense against herbivores - PubMed (nih.gov). There may be more examples that are relevant to include.

Figure S12. A bootstrap value of 42 means that in 58% of the pseudoreplicates the branch was elsewhere. Generally bootstrap values below 70 should not be displayed

Some enzymes/genes are discussed but without introduction, eg F6'H CL.

Some of especially the supplementary tables but also some the supplementary files are not easy to follow. Perhaps adding a brief explanation would help the reader.

Reviewer #1 (Remarks to the Author):

In the revised manuscript, my comments were satisfactorily answered. The current revision is well organized and clearer to follow. Regarding the novelty in this work, it still did not meet my expectations. But given the massive amount of work and well-revised manuscript, it should be encouraged. I recommend this report for publication in Nature Communications.

Response: We appreciate your positive feedback and thank you for helping to improve our manuscript.

Reviewer #2 (Remarks to the Author):

I am fully satisfied with the revisions made by the authors. I have no further suggestions.

Response: We appreciate your positive feedback and thank you for helping to improve our manuscript.

Reviewer #3 (Remarks to the Author):

1. Regarding figure 1. Use both trivial names and show structures to combine figures 1 A, B, and C so its easier for the reader to follow. The notation that Escin is “in the seeds” is rather unclear. Could the authors be more clear about which tissues in the seed?

Response: Thank you very much for your suggestion. We have revised Figure 1 to present the aescin distribution in the fruit seeds more clearly.

2. Two CYP 72As are mentioned in the text; however there are more, eg CYP72A555 is involved in biosynthesis of saponins and CYP72s are involved I hydroxylation of gibberellins – suggesting that they have been recruited from that pathway. CYP72A enzymes catalyse 13-hydrolyzation of gibberellins - PubMed (nih.gov) and The cytochrome P450 CYP72A552 is key to production of hederagenin-based saponins that mediate plant defense against herbivores - PubMed (nih.gov). There may be more examples that are relevant to include.

Response: Thank you very much for your suggestion. In the revised manuscript, we have included a summary of the CYP72 family members involved in triterpenoid biosynthesis, as mentioned in lines 248-251. However, upon further investigation of CYP72A555, we could not find literature 51 that describes its catalytic function. Specifically, we added this sentence. “CYP72A members exhibited diverse hydroxylation activities towards subtract oleanane-type triterpenoids, including the C-30 oxidation of β -amyirin^{55,56}, C-21 oxidation of soyasapogenol B and β -amyirin⁴⁵. C-20 hydrodation of 13 β -epoxy and 16 β -hydroxy- β -amyirin³⁸, C-22 oxidation of 24-hydroxy- β -amyirin⁵⁷, and C-23⁵⁸ or C-2 β oxidation of oleanolic acid^{59,60}.”

45. Yano, R. *et al.* Metabolic switching of astringent and beneficial triterpenoid saponins in soybean is achieved by a loss-of-function mutation in cytochrome P450 72A69. *Plant J.* **89**, 527–539 (2017).

55. Seki, H. et al. Triterpene functional genomics in licorice for identification of CYP72A154 involved in the biosynthesis of glycyrrhizin. *Plant Cell* 23, 4112–4123 (2011).
56. Fanani, M. Z. et al. Molecular Basis of C-30 Product Regioselectivity of Legume Oxidases Involved in High-Value Triterpenoid Biosynthesis. *Front. Plant Sci.* 10, 1–16 (2019).
57. Fukushima, E. O. et al. Combinatorial biosynthesis of legume natural and rare triterpenoids in engineered yeast. *Plant Cell Physiol.* 54, 740–749 (2013).
58. Liu, Q. et al. The cytochrome P450 CYP72A552 is key to production of hederagenin-based saponins that mediate plant defense against herbivores. *New Phytol.* 222, 1599–1609 (2019).
59. Wang, Y. et al. Deletion and tandem duplications of biosynthetic genes drive the diversity of triterpenoids in *Aralia elata*. *Nat. Commun.* 13, (2022).
60. Biazzi, E. et al. CYP72A67 catalyzes a key oxidative step in *Medicago truncatula* hemolytic saponin biosynthesis. *Mol. Plant* 8, 1493–1506 (2015).

3. Figure S12. A bootstrap value of 42 means that in 58% of the pseudoreplicates the branch was elsewhere. Generally bootstrap values below 70 should not be displayed

Response: Following your comment, we have revised this phylogenetic tree, and the numbers on the branch represented nodes with a support value greater than 70%.

4. Some enzymes/genes are discussed but without introduction, eg F6'H CL.

Response: Thank you for your suggestion. We have added a brief introduction of AcF6'H and Ac4CL functions in the introduction section.

5. Some of especially the supplementary tables but also some the supplementary files are not easy to follow. Perhaps adding a brief explanation would help the reader.

Response: Thank you very much for your advice. To help readers understand the Supplemental Files, we have created a catalog for all Supplemental Figures and Tables.

Reviewers' Comments:

Reviewer #3:

Remarks to the Author:

Thank you for the revised manuscript.

I am not a seed anatomically expert, but I still find the term "seeds" anatomically rather loose. Is the signal in the endosperm and as shown in the figure the signal/distribution does not seem to be uniformly distributed in the seed?

The other paper, besides CYP72A552, that I mentioned were the CYP72As that catalyze C13 hydroxylation of not triterpenoids but gibberellin phytohormones.

Nat Plants. 2019 Oct;5(10):1057-1065. doi: 10.1038/s41477-019-0511-z. Epub 2019 Sep 16.

CYP72A enzymes catalyze 13-hydroxylation of gibberellins.

In figure FS15 which bootstraps is referred to in the figure?

Reviewer #3 (Remarks to the Author):

Thank you for the revised manuscript.

1. I am not a seed anatomically expert, but I still find the term “seeds” anatomically rather loose. Is the signal in the endosperm and as shown in the figure the signal/distribution does not seem to be uniformly distributed in the seed?

Response: Thank you very much for your suggestion. According to the relevant literature ^[1] and the Flora of China ^[2], the *Aesculus* capsule can be manually split into pericarp and seed. Figure 1C illustrates the anatomical structure of the capsule. Following your comment, we have made revisions to Figure 1 and the main text (Lines 90-99). The analytical results showed escin Ia and its isomer escin Ib (m/z 1169.5152 [$C_{55}H_{86}O_{24}+K$]⁺) massively accumulated in the cotyledons near the testas (Figures 1 C, Table S3).

[1]. List A, Steward FC. The nucellus, embryo sac, endosperm, and embryo of *Aesculus* and their interdependence during growth. *Ann Bot.* 1965;29:1–15.

[2] Flora of China, http://www.efloras.org/florataxon.aspx?flora_id=2&taxon_id=100706

2. The other paper, besides CYP72A552, that I mentioned were the CYP72As that catalyze C13 hydroxylation of not triterpenoids but gibberellin phytohormones.

Nat Plants. 2019 Oct;5(10):1057-1065. doi: 10.1038/s41477-019-0511-z. Epub 2019 Sep 16.

CYP72A enzymes catalyze 13-hydrolyzation of gibberellins.

Response: Thank you very much for your suggestion. We added the description of CYP72A, which hydrolyzes gibberellins, in the main text (Lines 246-247).

3. In figure FS15 which bootstraps is referred to in the figure?

Response: Following your comment, we have revised the phylogenetic tree of Figure S15B. Branch support values greater than 70% are now presented. In addition, we revised Figure S21 to include the bootstrap value of the BAHD phylogenetic tree.

Reviewers' Comments:

Reviewer #3:

Remarks to the Author:

The authors have responded well to my last comments.